# Apitherapy for Age-Related Skeletal Muscle Dysfunction (Sarcopenia): A Review on the Effects of Royal Jelly, Propolis, and Bee Pollen

**DOI:** 10.3390/foods9101362

**Published:** 2020-09-25

**Authors:** Amira Mohammed Ali, Hiroshi Kunugi

**Affiliations:** 1Department of Mental Disorder Research, National Institute of Neuroscience, National Center of Neurology and Psychiatry, Tokyo 187-0031, Japan; hkunugi@ncnp.go.jp; 2Department of Psychiatric Nursing and Mental Health, Faculty of Nursing, Alexandria University, Alexandria 21527, Egypt; 3Department of Psychiatry, Teikyo University School of Medicine, Tokyo 173-8605, Japan

**Keywords:** apitherapy, royal jelly, propolis, bee pollen, sarcopenia, dietary interventions, muscle, skeletal, muscle wasting, physical performance, coronavirus disease 2019, COVID-19, body composition, lean body mass, insulin resistance, mitochondrial dysfunction, satellite stem cells

## Abstract

The global pandemic of sarcopenia, skeletal muscle loss and weakness, which prevails in up to 50% of older adults is increasing worldwide due to the expansion of aging populations. It is now striking young and midlife adults as well because of sedentary lifestyle and increased intake of unhealthy food (e.g., western diet). The lockdown measures and economic turndown associated with the current outbreak of Coronavirus Disease 2019 (COVID-19) are likely to increase the prevalence of sarcopenia by promoting sedentarism and unhealthy patterns of eating. Sarcopenia has multiple detrimental effects including falls, hospitalization, disability, and institutionalization. Although a few pharmacological agents (e.g., bimagrumab, sarconeos, and exercise mimetics) are being explored in different stages of trials, not a single drug has been approved for sarcopenia treatment. Hence, research has focused on testing the effect of nutraceuticals, such as bee products, as safe treatments to prevent and/or treat sarcopenia. Royal jelly, propolis, and bee pollen are common bee products that are rich in highly potent antioxidants such as flavonoids, phenols, and amino acids. These products, in order, stimulate larval development into queen bees, promote defenses of the bee hive against microbial and environmental threats, and increase royal jelly production by nurse bees. Thanks to their versatile pharmacological activities (e.g., anti-aging, anti-inflammatory, anticarcinogenic, antimicrobial, etc.), these products have been used to treat multiple chronic conditions that predispose to muscle wasting such as hypertension, diabetes mellitus, cardiovascular disorder, and cancer, to name a few. They were also used in some evolving studies to treat sarcopenia in laboratory animals and, to a limited degree, in humans. However, a collective understanding of the effect and mechanism of action of these products in skeletal muscle is not well-developed. Therefore, this review examines the literature for possible effects of royal jelly, bee pollen, and propolis on skeletal muscle in aged experimental models, muscle cell cultures, and humans. Collectively, data from reviewed studies denote varying levels of positive effects of bee products on muscle mass, strength, and function. The likely underlying mechanisms include amelioration of inflammation and oxidative damages, promotion of metabolic regulation, enhancement of satellite stem cell responsiveness, improvement of muscular blood supply, inhibition of catabolic genes, and promotion of peripheral neuronal regeneration. This review offers suggestions for other mechanisms to be explored and provides guidance for future trials investigating the effects of bee products among people with sarcopenia.

## 1. Introduction

While the numbers of older adults are expanding all over the world, the pandemic of sarcopenia, skeletal muscle loss/weakness is also on the rise [1]. Loss of lean body mass is a direct effect of the inflammatory and oxidative conditions that develop with aging [1,2]—described as inflammaging. Both inflammatory mediators and free radicals, which are highly expressed in older seniors and in patients with chronic diseases, such as type 2 diabetes mellitus, cause remarkable shrinkage of fast-twitch type II fibers promoting their transformation into the slow-twitch type I fibers [1,3,4,5].

Research indicates that muscle wasting may develop after adolescence (at the beginning of the third decade of life) as a result of sedentary lifestyle and improper diet (low protein/high fat/low fiber) [6,7,8]. These behavioral factors alter the composition of gut microbiota promoting gut dysbiosis, which allows the passage of bacterial endotoxins into the systemic circulation to induce inflammation and oxidative stress same as in immunological aging that occurs during old age [9,10].

Longitudinal data show that the currently-occurring global crisis, coronavirus disease 2019 (COVID-19), is likely to aggravate the development of sarcopenia in young groups by promoting unhealthy lifestyle [11]. On one hand, physical inactivity is increasing as a result of the lockdown measures and stay home strategies adopted by most governments to limit the spread of this infection [11,12,13]. On the other hand, food production has been seriously affected during the COVID-19 outbreak along with increase in food prices, which prompt people to consume unhealthy/processed foods since they are cheaper than fresh and healthy ones [14].

Given its high prevalence in the general population, which ranges between 5 and 40% in western countries and increases up to 50% in advanced age, sarcopenia is considered a public health problem [4,15]. It causes progressive decline of functional capacity, contributes to frailty, increases the risk of falls and hospitalization in old people [16,17], and leads to poor outcomes in patients with COVID-19 [18]. A large number of pharmacological agents are being tested as anti-sarcopenic agents such as bimagrumab (BYM338), enobasarm (GTx-024), trevogrumab (REGN1033), and sarconeos (BIO101). Most trials are in phase 1 or phase 2 [19]. In addition, outcomes of commonly used treatments (e.g., testosterone, growth hormone, and anabolic steroids) have been rather unsatisfactory [4,20,21,22]. Therefore, the most appropriate strategies for preventing and treating sarcopenia are limited to physical exercise and protein-rich diet [6,21]. Nevertheless, old people tend to be less compliant with physical activity programs [23,24]. Meanwhile, problems of the gastrointestinal (GI) tract that develop during advanced age (e.g., loss of teeth, taste, smell, and decreased absorption) as well as anabolic resistance of aged muscle limit potential benefits of protein-rich food in this group [25,26]. Therefore, it is of importance to search for novel preventive and curative strategies for sarcopenia, which can take into account the multifactorial nature of aging-related skeletal muscle failure.

## 2. Apitherapy as a Possible Complementary Treatment for Sarcopenia

Rock paintings from the Stone Age portray consumption of bee products by humans [27]. The first evidence of human usage of bee products for therapeutic and cosmetic purposes dates back 6000 years in ancient Egypt and later in China, Greece, and Rome [27,28,29,30]. Current research interest is directed toward the use of natural substances, including bee products, as potential pharmaceuticals to modify disease progression [31]. The term “apitherapy” describes a category of complementary and alternative medicine that comprises therapeutic use of various bee products including apilarnil (atomized drone larva) to prevent and treat illnesses [30].

Bee workers of either *Apis mellifera* or *Apis cerana* species—the former is common in Europe, Asia, Africa, and America while the latter exists only in southern and southeastern Asia—produce and store multiple bioactive substances [32]. Royal jelly, propolis, bee pollen, honey, bee venom, bee bread, and bee wax are common products of the bee hive. They all (to a varying degree) possess multiple health promoting properties due to their high content of natural antioxidants such as flavonoids, phenols, or terpenoids [28,32]. Research documents variability in contents and effects of every single bee product, mainly due to the influence of bee species, botanical origin, geographic location, season, extraction, and handling procedures [2,28].

Several lines of evidence describe anti-aging effects of royal jelly, bee pollen, and propolis both in humans and laboratory animals [2,33,34,35]. These three products are widely used as dietary supplements [36,37,38,39]. In the meantime, the literature gives examples of numerous dietary supplements that could successfully prevent or alleviate the progression of muscle mass loss in old age [40,41,42]. Bee products represent a part of this interventional strategy. However, the extent to which bee products can affect sarcopenia as well as understanding of their underlying mechanism of action are far from being clear. Therefore, we conducted this review with the aim of investigating the anti-aging properties of these products with a focus on skeletal muscle functioning in advanced age. In this respect, we reviewed animal and human studies investigating effects of the aforementioned products on skeletal muscle aging and elaborated on different mechanisms underlying these effects. Studies included in this review were retrieved by searching PubMed and Google scholar using a combination of terminologies of “sarcopenia, muscle wasting, muscle mass, lean body mass, skeletal muscle, motor” with “royal jelly, honey, bee pollen, propolis, bee venom, bee bread, bee wax, chrysin, apamin, caffeic acid phenethyl ester”. Snow ball manual search using reference lists of retrieved studies was also conducted. This search resulted in a number of studies, which addressed muscle wasting and related dynamics through the use of three bee products, namely royal jelly, bee pollen, and propolis. Figure 1, Panel A and Panel B, summarizes the chemical composition and biological properties of these bee products while this section elaborates on these products in depth.

### 2.1. Royal Jelly: Its Constituents, Biological, and Pharmacological Activities

Royal jelly is a thick, milky, white-yellowish, acidic colloid substance secreted from the hypopharyngeal and mandibular salivary glands of young nurse honey bees (5–15 days old) [32,43]. In general, fresh royal jelly mostly consists of water (67% *w*/*w*) in addition to carbohydrates (16%), proteins and amino acids (12.5%), fat (5%), and many other elements [32]. However, royal jelly content of these substances noticeably varies depending on numerous factors like botanical source, bee species, bee artificial feeding, weather, season, location, method of processing, and the like [2,44].

Protein is the most copious active element in royal jelly, representing half the weight of its dry matter [2]. It vastly comprises nine 49–87 kDa water-insoluble proteins, known as major royal jelly proteins 1–9 (MRJPs1-9) [2,45]. MRJPs constitute more than 80% of royal jelly protein content, and MRJPs1–5 constitute 82–90% of all MRJPs. MRJPs contain 400–578 amino acids that contribute to the antioxidant effect of royal jelly as well as its role in cell proliferation, cell adhesion, cell growth, and immunity [46,47]. Novel non-MRJPs proteins have been newly discovered [48]. Royalisin, jelleines, and aspimin are examples of other proteins that exist in royal jelly, albeit in small amounts. These proteins as well as MRJPs demonstrate strong antimicrobial and bactericidal activities even against the most drug-resistant bacterial strains such as methicillin-resistant *Staphylococcus aureus*, *Pseudomonas aeruginosa*, *Klebsiella pneumoniae*, vancomycin-resistant *Enterococci*, as well as extended-spectrum β-lactamase-producing *Proteus mirabilis* and *Escherichia coli* [28,29]. 

Carbohydrates (e.g., fructose, glucose maltose, trehalose, melibiose, ribose, and erlose) constitute 7.5–16% or royal jelly content [49]. Reducing sugars in royal jelly are thought to contribute to its epigenetic effect through the activation of insulin-like growth factor 1 (IGF-1) and mammalian target of rapamycin (mTOR) signaling cascades. Thus, they stimulate caste differentiation of *Apis mellifera* larvae into queens by increasing intake of food and key nutrients [50].

Lipids make up 7–18% of the dry weight of royal jelly. This fraction largely comprises a group of unique and rare saturated or monounsaturated short and medium chain fatty acids that are terminally or internally hydroxylated with terminal mono- or dicarboxylic acid functions [2,28]. The vast majority of royal jelly fat content (80–85%) consists of short hydroxyl fatty acids such as trans-10-hydroxy-2-decenoic acid (10-HDA), which exists only in royal jelly; and therefore, it is known as royal jelly acid or queen bee acid [28,49,51]. 10-HDA is one of the most potent bioactive elements in royal jelly expressing strong anti-aging, neuroprotective, antiproliferative, antimicrobial, antioxidant, anti-inflammatory, and epigenetic effects [52,53,54,55,56,57,58]. In addition, the lipid fraction of royal jelly contains phenolic acids (4–10%), wax (5–6%), steroids (3–4%), and phospholipids (0.4–0.8%) [49]. 

A wide range of minor constituents and bioactive compounds exist profusely in royal jelly such as acetylcholine, nucleotides (adenosine, guanosine, adenosine tri-phosphate (ATP), adenosine monophosphate (AMP)), minerals (iron, sodium, calcium, potassium, zinc, magnesium, manganese, and copper), amino acids (8 out of 9 essential amino acids Val, Leu, Ile, Thr, Met, Phe, Lys, and Trp), vitamins (retinol (A), ascorbic acid (C), tocopherol (E), riboflavin (B2), niacin (B3), and other B vitamins), esters, aldehydes, ketones, alcohol, and minor heterocyclic compounds [2,28,49,59,60,61]. It is worth noting that royal jelly loses most of its bioactive ingredients and biological properties when stored at a temperature of 5 °C or higher. Therefore, freezing is the best method to store royal jelly [62]. Enzymatic treatment of royal jelly removes allergen proteins and enhances its nutrient content in addition to improving its digestibility and absorption in the gut without altering its freshness [2,59].

Royal jelly has been historically used as a beautifying agent by famous queens such as Cleopatra, and it is still involved in the cosmetic industry [29,56]. Its rich content of bioactive compounds grants it a plethora of diverse health benefits such as antioxidant, anti-inflammatory, neurotrophic, hypotensive, antidiabetic, antilipidemic, antirheumatic, anticarcinogenic, anti-fatigue, antiadipogenic, and antimicrobial activities [43,45,63]. Therefore, it is widely used to treat multiple serious conditions including diabetes, hypertension, hyperlipidemia, cancer, skin diseases, and neurodegenerative diseases such as Alzheimer’s disease and Parkinson’s disease [2,43,46,59,64]. In addition, bee queens (which enjoy long lifespan as well as super fertility and physical qualities) consume royal jelly throughout their entire lives, and royal jelly is considered a promising anti-aging nutraceutical that can positively enhance fertility and improve body composition [2]. 

### 2.2. Propolis: Its Constituents, Biological, and Pharmacological Activities

Propolis, also known as bee glue, is a sticky wax-like substance that constitutes a mixture of bee salivary secretions, bee wax, and resinous sap occurring in the bark and leaf-buds of specific plants [37,65]. It comes in green, red, brown, or black colors based on the collected local flora [66]. The word propolis comprises two Greek words “pro” and “polis”, which in order mean “in front of or at the entrance to” and “community or city”. Propolis is a hive-defensive substance, which bees use to protect and repair their hives [67].

Propolis is a unique product of a complex composition that comprises more than 420 chemical substances [37,68]. Nonetheless, its composition and biological activities vary considerably depending on its botanical and geographical origins as well as the time of harvesting [38,65,67]. Propolis is rich in oxyprenylated phenylpropanoids—secondary metabolites from plants, fungi, and bacteria [69]—such as 7-isopentenyloxucoumarin, boropinic acid, 4-geranyloxyferulic acid, and auraptene. The last two exist in raw Italian propolis at high concentrations: 107.12 and 145.37 μg/g of dry propolis, respectively. Flavonoids, a large group of phenolic compounds, are abundant in Italian propolis, and they are differentiated into several groups including flavanones (e.g., naringenine, 4.4 mg/g), flavones (e.g., apigenine, 1.7 mg/g), flavonols (e.g., galaning, 0.9 mg/g), tannins (e.g., gallic acid 8.4 mg/g), catechins (expressed as (+)-catechin 0.4 mg/g, and caffeic acid and its esters (expressed as caffeic acid, 9.2 mg/g) [69]. The most profuse flavonoids in ethanolic extracts of Brazilian propolis are artepillin C (38.6 mg/g), coumaric acid (10.6 mg/g), and kaempferide (12.6 mg/g) [70]. Key other constituents of propolis include polyphenol (e.g., phenolic acids and aromatic esters), phenolic aldehydes, terpenoids, ketones, enzymes (e.g., α- and β-amylase), vitamins (e.g., thiamin (B1), riboflavin (B2), pyridoxine (B6), ascorbic acid (C), tocopherol (E)), minerals (e.g., calcium, potassium, magnesium, iron, sodium, barium) essential oils, alcohol, fatty acids, β-steroids, and many other elements [37,38,67,68,71].

The attention of several drug targeting studies has recently been focused on the therapeutic activities of individual bioactive compounds in propolis [65,68]. Flavonoids comprise the majority of mostly studies bioactive substances in propolis. Chrysin (5,7-dihydroxyflavone) is a flavonoid that exists in certain mushrooms, flowers (e.g., blue passion flower), and in other bee products (e.g., honey). It expresses anti-inflammatory, antioxidant, anti-proliferative, and neuroprotective effects [72]. Caffeic acid phenethyl ester (CAPE), a derivative of hydroxycinnamic acid, expresses anti-oxidant, immunomodulatory, anti-inflammatory, antiviral, and ant-neoplastic properties [73,74,75]. Pinocembrin (5,7-dihydroxyflavanone) is the most copious flavonoid in propolis—1 g of balsam/an ethanolic extract from poplar propolis found in Spain contains up to 606–701 mg of pinocembrin [76]. It exists in numerous plants (e.g., Eucalyptus and Populus). It exhibits anti-inflammatory, antioxidant, antimicrobial, and antiproliferative activities [77,78].

Essential/volatile oils are major bioactive constituents of propolis, and they contribute to its special aroma [79,80]. They also, partially, contribute to the strong antimicrobial, antioxidant, and anticancer activities of propolis [79,81,82]. The volatile fraction of propolis varies in each sample even within a single country due to plant source and climate [79]. For instance, cumulative knowledge shows that volatile oils in propolis found in countries surrounding the Mediterranean depend mainly on the botanical origin. They primarily comprise poplar-derived compounds (e.g., benzoic acid and its esters and oxygenated sesquiterpene β-eudesmol) and conifer-derived compounds such as the hydrocarbon monoterpene α-pinene [80]. Interestingly, the number of volatile compounds derived from a single type of propolis is also reported to vary according to extraction techniques. In this regard, reports from China show that traditional hydrodistillation, steam-distillation extraction, and dynamic headspace sampling could characterize around 12, 40 and 70 type of volatile components of propolis, respectively [79]. Moreover, the level of antimicrobial activity of volatile compounds of propolis greatly depends on their extent of purification [82].

Thanks to its countless bioactive elements, propolis enjoys a range of versatile biological and pharmacological properties including antimicrobial, antiviral, antifungal, antioxidant, anti-inflammatory, antineoplastic, antiaging, and cytostatic properties. In addition, it is considered a perfect natural food preservative due to its antimicrobial activity [35,38,65,66,68,71]. Because of its enormous health-promoting activities, propolis is widely used as a dietary supplement in many countries, especially in Japan [37,38,39].

Propolis is not suitable for use in its crude state since it may contact harmful materials e.g., asphalt from the road [68]. Using solvents like ethanol, glycerol, chloroform, ether and acetone or water is necessary to get rid of hazardous substances and to increase its yield of bioactive compounds [67,68]. Although water may be a cheap solvent, propolis has poor solubility in water. Therefore, propolis water extracts are 10-fold lower in their phenolic contents than ethanol extracts. In addition, they retain the strong flavor and aroma of propolis [68]. Moreover, propolis contains allergenic components: caffeic acids derivates (e.g., 3-methyl-2-butenyl caffeate and phenylethyl caffeate), as well as benzyl salicylate and benzyl cinnamate [80]. Therefore, propolis use/consumption should be contraindicated in individuals with known allergies. 

### 2.3. Bee Pollen: Its Constituents, Biological, and Pharmacological Activities

Bee pollen is an api-material that originally comprises male gametophytes or spermatophytes of flowers, which stick to bee body. Bee workers mix these floral pollens with honey, nectar, and bee saliva. The latter is rich in various enzymes e.g., amylase, catalase, as well as lactic acid bacteria, which cause pollen fermentation [36,83,84]. Hence, the tiny wind pollen grains collected by bees aggregate together to form granules or pellets of 1.4–4 mm in size [84]. 

In addition to water, which in order constitutes 20–30% and 6–8% of the content of recently collected and dried bee pollen, bee pollen contains around 200 chemical compounds. Like other bee products, its composition varies considerably according to botanical origin. Carbohydrates account for the most abundant ingredient (18.50–84.25%), and reducing sugars such as glucose, fructose, and sucrose constitute the vastest majority (13–55%). Other major elements include proteins and essential amino acids (5–60%), unsaturated and saturated fatty acids (0.15–31.26%), crude fiber (0.3–20%), nucleic acids (especially RNA), and various minerals (e.g., potassium, phosphorus, calcium, magnesium, zinc, copper, manganese, and iron) [36,83,84,85,86]. In addition, its average total phenolic content is 30.59 mg gallic acid equivalents (GAE)/g, but again it varies considerably based on floral origin (0.69–213.20 mg GAE/g) [86]. Moreover, bee pollen is abundant in both water- and fat-soluble vitamins e.g., β-carotene (vitamin A precursor), ascorbic acid (C), tocopherol (E), folic acid (B9), and other vitamin B, especially niacin. Bee pollen contains other elements that still need to be explored (2–5%) [83]. Therefore, bee pollen represents a perfect whole health-promoting food. In fact, comparisons of the percentages of nutrients in bee pollen with daily required intake of an adult individual revealed that few grams of bee pollen can meet daily human nutritional requirements [83,84].

Bee pollen demonstrates various biological properties and therapeutic activities e.g., antioxidant, anti-inflammatory, anti-lipidemic, anticancer, antiallergic, and antimicrobial [36,87]. Existing knowledge emphasizes its antiaging effects: it reduced the production of age-related pigment known as lipofuscin (induced by oral peroxidized corn oil or intravenous alloxan injection) in cardiac muscle, brain, liver, and suprarenal gland in aged mice (reviewed in [34]).

The composition of bee pollens depends primarily on its botanical source since nutrient contents (e.g., polyphenols) of pollen grains, which support their survival and fusion with female gametes, vary between different plants [83,84]. Storage conditions are of great importance were it to retain its biological activities. Bee pollen should be consumed fresh soon after collection. Most of its major elements (reducing sugars, total proteins, vitamin C, and provitamin A) are destroyed at 40 °C. Lyophilization damages its vitamin content while freezing is recommended for the storage of bee pollen since it does not affect its chemical structure [83].

Dry pollen pellets resist decay due to their tough outer coat, which comprises two layers made of cellulose and sporopollenin [88,89]. However, ingestion of bee pollen by humans may not yield its optimal nutritional value because the hard sporopollenin shell hinders access of digestive secretions to the nutrient-rich core of the pellet. Biological, chemical, and mechanical techniques are used to break bee pollen microcapsules in order to enhance its digestibility in the gut. However, these methods may be expensive or ineffective i.e., they degrade important nutrients via enzymatic activity [88,90]. Ultraviolet spectroscopy and high performance liquid chromatography-photo diode array show that processing bee pollen through the use of an edible lipid-surfactant mixture (Captex 355 and Tween 80) increases its yield of polyphenols and flavonoid aglycones [90].

### 2.4. Safety Profile of Royal Jelly, Propolis, and Bee Pollen

Propolis exists in a plethora of commercial products that are directly consumed or used by humans e.g., lozenges, soap, toothpastes and mouth wash, creams, gels, cough syrups, wines, cakes, chewing gums, candies, shampoo, chocolate, skin lotions, processed meat, etc. [67]. In addition, royal jelly, bee pollen, and propolis are widely used as dietary supplements in many parts of the world [36,37,38,39]. Existing knowledge denotes no adverse effects from their consumption either in rodents or in humans [39,66]. The safety of pinocembrin, a flavonoid available in propolis and an approved drug in China, is documented since its elimination from the body is rapid [91]. The safety profile of bee pollen (both crude and processed) has been empirically tested. Oral consumption of bee pollen (up to 2 g/kg body weight) expressed no allergic reactions in mice including behavioral changes, salivation, diarrhea, respiratory or autonomic responses, restlessness, convulsions, tremors, or death [90]. In fact, the German Federal Board of Health acknowledges bee pollen as an official medicine [36].

Several lines of evidence support the anti-allergic effect of propolis and royal jelly. This effect involves inhibiting mast cell degranulation, suppressing cysteinyl-leukotriene release, as well as reducing serum histamine, IgG, and IgE levels in various allergic conditions by suppressing histamine H1 receptor [37,39,92]. Nevertheless, rare allergic reactions to bee products other than bee venom are documented in the literature. They are most frequent in small children [80,93]. Examples of such reactions comprise contact dermatitis in beekeepers following the handling of propolis, as well as contact stomatitis and oral mucositis after the usage of lozenges containing propolis [80]. Hence, bee products should be used with caution, especially in people with known allergies, pregnant and lactating women, and small children [61]. In addition, bee products can be safely consumed after adequate processing. Processing involves removal of known allergens such as enzyme treatment of royal jelly and filtration of bee venom by stepped-gradient open column [2,94].

## 3. Evidence of Anti-Sarcopenia Effects of Bee Products from Preclinical and Clinical Studies

Though few animal models were used to examine the effect of bee products on sarcopenia, findings indicate that royal jelly, propolis, and bee pollen can induce both structural and symptomatic improvements and reduce behavioral dysfunctions associated with sarcopenia in rodents (Figure 1, Panel C). Both crude and protease-treated royal jelly (pRJ) significantly delayed age-related impairment of motor functions in d-galactose induced mouse model of aging [95], naturally aged sarcopenic mice [96], and in genetically heterogeneous head tilt (HET) mice—which exhibit vestibular dysfunction, imbalanced position, and inability to swim—by improving performance on grip strength, wire hang, horizontal bar, and rotarod tests [97]. Similarly, royal jelly improved physical performance in aged rodents—it significantly increased the number of crossings and swimming speed and prolonged swimming distance in water maze [96,98,99]. In addition, royal jelly decreased lipid accumulation in skeletal muscle [100], positively improved the size of muscle fibers, lowered age-related reduction of skeletal muscle weight [95,96,97,100], increased the differentiation and proliferation rate of muscle satellite cell, improved the regenerative capacity of injured muscle, and suppressed catabolic genes in aged mice with sarcopenia and in HET mice [96,97]. The muscle mass-accelerating effects of 10-HDA, a key fatty acid in royal jelly, were more pronounced in male animals than in females. However, 10-HDA mitigated the accumulation of adipose tissue in female mice [54]. It is note-worthy that pRJ had no effect on muscle strength and physical performance in humans aged 70 years and above [59].

The effects of other bee products on muscle mass were mostly positive. Bee pollen promoted body weight regain and increased the relative weight of the gastrocnemius muscle in eccentric exercising rats [90]. It also increased the absolute weights of plantaris and gastrocnemius muscles in malnourished old rats [36]. CAPE restored gastrocnemius muscle mass in rats on exhaustive exercise and in rats with ischemia reperfusion [101,102,103]. A study reported no effect of propolis ethanolic extracts (4% of diet) on the size of muscle or their level of myostatin in Nile tilapia. However, another interesting study reported significant increases in body protein deposition and body condition factor—an estimate of future growth, survival, and reproductive potential—in Nile tilapia post-larvae and fingerlings receiving 2.6 g propolis/kg of feed [104]. Nonetheless, these findings seem to be bound to fish. Interestingly, supplementing obese rats on high fat diet (HFD) with milk naturally enriched with long-chain polyunsaturated fatty acids (PUFA) and polyphenols from propolis significantly increased gastrocnemius muscle mass compared with whole milk and milk enriched with PUFA only [105]. Several molecular changes were associated with these effects. We elaborate on these changes in Section 4. Table 1 presents more details on treatments with bee products and key findings of the relevant studies.

## 4. Mechanisms of Action of Royal Jelly, Bee Pollen, and Propolis in Sarcopenia

Relatively few studies have explored the mechanism through which royal jelly, bee pollen, and propolis may be beneficial for sarcopenia. In vivo and in vitro studies included in this review uncovered a number of inter-related cellular and molecular events that underlie the effect of these bee products on skeletal muscle including suppression of catabolic genes, counteracting metabolic abnormalities, inflammation, and oxidative damages as well as enhancement of motor neuronal regeneration, promotion of stem cell function, and correction of the structure of gut microbiome (Figure 1, Panel D). These mechanisms are exclusively described in the coming paragraphs.

### 4.1. Modulating Inflammatory Responses in Skeletal Muscle

The role that inflammation plays in skeletal muscle is not quite clear as it looks. Inflammatory mediators behave in a dual fashion in muscle cells. During injury, cytokines and chemokines stimulate muscle repair and regeneration via activation of myoblasts, a core event in muscle remodeling [108]. Similarly, serum and muscle levels of IL-6 temporarily increase following physical exercise, and IL-6 blocks the activity of catabolic cytokines such as TNF-α. On the other hand, chronic inflammation in muscle cells, which correlates with persistent mitochondrial dysfunction and metabolic dysregulation, pathologically activates muscle fiber transformation and atrophy, eventually resulting in the development of sarcopenia [23,114]. It seems that bee products also act dually in skeletal muscle: they support the activity of cytokines that promote muscle remodeling [108] and suppress muscle-consuming cytokines [38,101,108]. In this regard, treating undifferentiated C2C12 myoblasts with ethanolic extracts of Brazilian propolis (100 μg/mL for 8 h) triggered the migration of RAW264 macrophage and increased their production of angiogenic factors (e.g., vascular endothelial growth factor A (VEGF-A) and metalloproteinase-12 (MMP-12)), chemokines (e.g., CCL-2 and CCL-5), and cytokines (e.g., IL-6, which increased by 40-folds). Propolis inhibited the expression of IL-1β and TNF-α at 4, 8, and 12 h of incubation. These effects were nuclear factor kappa B (NF-κB)-dependent given that propolis simultaneously increased nuclear translocation of p65 and p50 NF-κB proteins 3 h after treatment. Meanwhile, inhibiting IκB kinase (IKK) by BMS-345541 profoundly hindered the effect of propolis on the expression of CCL-2, CCL-5, and IL-6 by 66%, 81%, and 69%, respectively. Propolis also enhanced the expression of MAIL/IκBζ. This molecule modulates chromatin and selectively induces the production of IL-6, leukemia inhibitory factor (LIF), and CCL-2 [108]. 

Chronic muscle tissue infiltration by inflammatory cells (e.g., leukocytes, neutrophils, and monocytes) activates oxidative and inflammatory signaling cascades that degrade cellular structures and promote necrosis such as inducible nitric oxide synthase (iNOS) and NF-kB [90,101,115]. CAPE and high levels (200 and 300 mg/kg) of crude and processed bee pollen reduced inflammatory cell infiltration into the gastrocnemius muscle of rats with muscle injury induced by eccentric exercise and ischemia reperfusion [90,101,102]. This effect was revealed by lower activity of myeloperoxidase, an indicator of neutrophil sequestration [101,102]. As such, CAPE and bee pollen not only suppressed lipid peroxidation (lower levels of malondialdehyde, MDA) but also inhibited the activity of myostatin and the production of muscle depleting cytokines and chemokines such as IL-1β, α2-macroglobulin, and monocyte chemotactic protein 1 (MCP-1) [36,38,90,101]. The underlying mechanism entailed downregulation of nuclear p65NF-κB and blockage of its consensus binding sites in skeletal muscle [101]. As a result, bee products increased muscle mass both in sarcopenic obese rat and malnourished old rats [36] and restored the structure of myofibers (despite the persistence of necrosis) compared with untreated eccentric exercising animals, which demonstrated necrotic and fragmented myofibers [90]. 

Royal jelly downregulated the activity of tumor necrosis factor receptor 1 (TNFR1) in the adipose tissue of aged obese rats receiving HFD [100]. TNFR1 interacts with TNFR2 to negatively regulate toll-like receptors (TLR) and Nod-like receptor signaling and stimulate excessive release of cytokines via activation of key inflammatory pathways such as NF-κB [107]. Mitigation of TNFR1 was associated with a significant increase of the weight of hind limb muscle and reductions in insulin levels, homeostatic model assessment of insulin resistance (HOMA-IR), serum lipids, muscle triglyceride levels, body weight gain, and abdominal fat weight. Therefore, royal jelly may protect against muscle loss in conditions involving impairment of the adipokine profile mainly through suppression of inflammatory responses associated with high fat mass, which is followed by correction of metabolic irregularities [100].

### 4.2. Counteracting Oxidative Stress in Skeletal Muscle

High production of reactive oxygen species (ROS) in skeletal muscle tissues has serious destructive effects, which alter the integrity of skeletal muscle resulting into fatigue, muscle wasting, and muscle weakness [90,101]. Sources of intramuscular ROS are numerous including mitochondrial dysfunction (e.g., alteration of mitochondrial enzymes in the respiratory chain as well as enzymes responsible for β-oxidation), neutrophil infiltration, and the activity of cytokines and major muscle degrading molecules such as myostatin [21,36,38,101,102,118]. Oxidative and nitrosative damages in skeletal muscle tissues are mediated by the activity of numerous pro-oxidant enzymes that are associated with inflammatory processes such as cyclooxygenase-2 (COX-2) and iNOS [101]. ROS triggers the activity of corrosive molecules such as poly (ADP-ribose) polymerase (PARP), xanthine oxidase, and adenosine deaminase, which contribute to DNA damage, lipid peroxidation (e.g., increased MDA), and protein nitrotyrosylation as well as ATP catabolism in muscle tissues [101,102]. 

Royal jelly enhanced the activity of antioxidant enzymes and suppressed lipid peroxidation in a d-galactose induced model of aging [95]. Two weeks of propolis treatment in rats undergoing hind limb unloading significantly reduced nuclear ROS levels and numbers of apoptotic endothelial cells in the soleus muscle to levels similar to normal rats [110]. Moreover, propolis significantly suppressed MDA activity in skeletal muscle and increased liver levels of SOD as well as gastrocnemius muscle levels of SOD, glutathione peroxidase and catalase in rats on eccentric exercise training (70% VO_2_max treadmill running exercise for 60 min) compared with rats receiving exercise alone or no treatment [109]. In addition, intraperitoneal pre-administration of CAPE (60 min before induction of ischemia reperfusion) significantly ameliorated the effects associated with high ROS levels, which accompany acute ischemia such as protein peroxidation and ATP catabolism in the gastrocnemius muscle [102]. 

Bee products probably reduced ROS production via regulation of the activity of mitochondrial enzymes. Royal jelly increased the maximal activity of citrate synthase (CS) and β-hydroxyacyl coenzyme A dehydrogenase (β-HAD) in the soleus muscle of rats on endurance training [106]. Bee pollen restored mitochondrial complex-I, -II, -III, and -IV enzyme activity to normal, increased SOD and glutathione, and reduced MDA, NO, and total protein content in the gastrocnemius muscle of rats on exhaustive exercise [90]. It also increased the activity of CS and complex IV in malnourished old rats via a mechanism that involved activation of mTOR [36]. mTOR is a major signaling pathway that regulates various signaling cascades involved in metabolism and autophagy such nuclear respiratory factor 2 (NRF2) [2]. Therefore, it is possible that the antioxidant activity demonstrated by bee products, particularly that expressed in the mitochondria, is associated with their metabolic and hypoglycemic activities. For instance, 10-HDA increased expression of peroxisome proliferator-activated receptor-γ coactivator-1α (PGC-1α) in skeletal muscle of diabetic mice [112,119]. Similarly, aged mice treated with amino acids similar to those found in royal jelly demonstrated improved size of muscle fiber by increasing PGC-1α mRNA levels. PGC-1α functions as a key regulator of sirtuin 1, which limits ROS production through stimulation of mitochondrial biogenesis and ROS defense system, thus protecting metabolically active tissues against oxidative damage [120]. Mechanistically, when PGC-1α gets activated, it interacts with other bioactive molecules such as muscle-specific transcription factors to stimulate the expression of genes that induce mitochondrial oxidative metabolism in brown fat and fiber-type switching in skeletal muscle [121].

The antioxidant activity of royal jelly and CAPE might be related to their strong capacity to activate the master redox-active NRF2 signaling pathway [73,122], which stimulates the production of internal antioxidants such as heme oxygenase-1 (HO-1), which scavenge free radicals [123]. Meanwhile, NRF2 and HO-1 block ROS production indirectly via suppression of inflammatory reactions [122]. In this context, CAPE reduced degenerative myopathy in rats on eccentric exercise via a complex mechanism that involved inhibition of NF-κB and its downstream pro-oxidant COX-2 and iNOS [101]. Correspondingly, CAPE decreased markers of oxidative cellular damages (protein carbonyl, protein nitrosylation, xanthine oxidase, and adenosine deaminase) associated with ischemia reperfusion and eccentric exercise in the gastrocnemius muscle [101,102,103]. In this regard, CAPE operated via a mechanism that involved inhibition of neutrophil and leukocyte infiltration into the gastrocnemius muscle, which was associated with decreased levels of myeloperoxidase. Myeloperoxidase contributes to excessive production of ROS and oxidative organ damage through a mechanism that embroils increased synthesis of hypochlorous acid [101,102]. Furthermore, CAPE accelerated purine salvage for ATP synthesis and inhibited ROS-induced lipid peroxidation via attenuation of the activity of adenosine deaminase [102]. 

In summary, the reported antioxidant effects of bee products were multifaceted involving increased production of antioxidant enzymes [90,95,109,110], and decreased ROS production [90,101,102,103,110] (secondary to reduction of inflammatory cell infiltration into skeletal muscle) [90,101,102,103], and restoration of mitochondrial activity [36,90,106].

### 4.3. Metabolic Regulation

Skeletal muscle is the main tissue that utilizes insulin for glucose uptake. Insulin regulates mitochondrial oxidative phosphorylation of proteins in human skeletal muscle and contributes to calcium mobilization from the sarcoplasmic-endoplasmic reticulum to mitochondria to stimulate the translocation of glucose transporter 4 (GLUT4) to the cell surface for glucose uptake. This process improves muscle protein synthesis in healthy people when the delivery of amino acids to skeletal muscle is increased, eventually leading to increased muscular mass [124,125,126]. Insulin resistance and glucose intolerance increase with old age evoking muscle loss. In this respect, hypoglycemic agents such as metformin can improve skeletal muscle metabolism via activation of adenosine monophosphate activated protein kinase (AMPK) [21].

AMPK, a heterotrimeric complex that consists of a catalytic subunit and two regulatory subunits, is an intracellular energy sensor that regulates glucose and lipid metabolism. It gets activated when cellular energy is depleted through allosteric binding of AMP or phosphorylation by AMPK kinase at Thr172 of the catalytic subunit by AMPK kinase. Upregulated AMPK activates signaling pathways that generate ATP from glucose and fatty acid oxidation, and it simultaneously blocks signaling that contributes to the synthesis of cholesterol, fatty acid, and triacylglycerol [111]. In addition, AMPK masters numerous signaling cascades such as Forkhead Box O transcription factor (FOXO) and AKT/mTOR, which regulate the expression of genes associated with inflammation, oxidative stress, mitochondrial function, autophagy, metabolism, and apoptosis [127,128].

The molecular events involved in the effect of bee products on catabolic genes and anabolic resistance in skeletal muscle could be much related to their hypoglycemic effects, which positively affect the quality of skeletal muscle. Evidence signifies a positive effect of royal jelly acid (10-HDA) on inflammation and autophagy via upregulation of AMPK, which subsequently alters NF-κB and NLRP3 inflammasome-IL1β signaling [129]. Positive effects of whole royal jelly on skeletal muscle are associated with improved insulin signaling [96,100]. In one study, royal jelly improved serum IGF-1 levels in aged rats and increased AKT signaling in satellite cells extracted from aged rats in a separate in vitro investigation [96]. In another study, royal jelly decreased fat mass and improved anabolic resistance in the skeletal muscle of old obese rats on HFD via downregulation of inflammatory responses in adipose tissue as indicated by downregulation of TNFR1. This effect was associated with enhanced sensitivity to insulin—portrayed by reduction of serum insulin level and HOMA-IR [100].

Japanese researchers proved that oral consumption of royal jelly and 10-HDA induced mitochondrial adaptation in the soleus muscle when accompanied with endurance training. These compounds also enhanced glucose uptake in skeletal muscle by inducing the phosphorylation of AMPK [106,112], an effect that was mediated by the upstream kinase Ca²⁺/calmodulin-dependent kinaseβ—independently of changes in AMP:ATP ratio and the liver kinase B1 pathway. Activation of AMPK was followed by translocation of GLUT4 to the plasma membrane of L6 myotubes [106]. It is note-worthy that effects of royal jelly on mitochondrial biogenesis under endurance training were muscle-specific. In this respect, neither endurance training nor royal jelly alone had an effect on the maximal activities of CS and β-HAD—the enzyme that catalyzes the rate-limiting step of β-oxidation of long-chain fatty acids—in the soleus muscle, which comprises type I fiber (around 35–45%) and type IIa (around 35–50%). On the other hand, royal jelly enhanced the activity of these enzymes in the soleus muscle of mice on endurance training. Of interest, endurance training increased the activity of CS and β-HAD in the plantaris and tibialis anterior muscles (which are mainly type II fiber with a total percentage of type IIb and type IIx fiber types of 90%) while royal jelly failed to exert an effect on these muscles in sedentary mice [106]. Nonetheless, the observed effects of royal jelly in the soleus muscle represent a merit. This is mainly because the oxidative type I fibers (e.g., soleus muscle) naturally undergo higher protein turnover (especially degradation), which makes them unable to grow in size or respond properly to insufficient nutrient intake [130].

Several lines of evidence indicate that propolis may affect muscle quality through the regulation of glucose metabolism [69,70,109,111]. This effect was vividly depicted in vivo by increased glycogen level in skeletal muscle and reduced serum levels of glucose and insulin [109]. Same as insulin, ethanolic extracts of propolis and CAPE induced glucose uptake [69,70,111] and potentiated insulin-mediated AKT activation and glucose uptake in differentiated L6 myoblast cells [111]. Likewise, Italian propolis at concentrations of 0.1 and 1 mg/mL as well as 4-geranyloxyferulic acid and auraptene (2 oxyprenylated phenylpropanoids, which are abundant in propolis) remarkably increased GLUT4 translocation to the plasma membrane and accelerated GLUT4-mediated glucose uptake in L6 skeletal myoblasts. The effect of propolis at a concentration of 11 mg/mL was significantly superior to the effect of insulin (0.1 μM), which was used as a positive control [69].

Similar to royal jelly, the effects of propolis (1 μg/mL), CAPE (10 μM), artepillin C, coumaric acid, and kaempferide on glucose metabolism occurred via activation of AMPK. These effects were comparable to those of 5-aminoimidazole-4-carboxamide ribonucleoside (AICAR), a potent AMPK activator. In the meantime, co-treatment with inhibitors of AMPK (e.g., compound C) and of phosphatidylinositol 3-kinase (PI3K) (e.g., LY-294002) blocked the effects of CAPE [70,111]. Phosphorylation of AMPK results in activation of the insulin receptor (IR) and subsequent phosphorylation of PI3K followed by activation of AKT and protein kinase C (PKC) leading to GLUT4 translocation and subsequent activation of several molecules that modulate insulin-stimulated glucose transport, eventually leading to glucose influx into cells of several tissues such as skeletal muscle and adipose tissue [69,70,111]. It is worth noting that the effects of CAPE on AMPK and AKT were quick (within 1 h and 3 min, respectively), and they vanished quickly (both molecules returned back to their basal levels within 12 h and 30 min, respectively) [111].

### 4.4. Enhancement of Muscle Protein Synthesis

Imbalance between muscle protein synthesis and degradation is associated with low protein intake and altered efficiency of the GI tract in old age, which triggers skeletal muscle loss and poor physical performance [36,40], given that proteins are the main building blocks of muscle myofibers. Moreover, transmembrane proteins and micro-peptides (e.g., myomixer and myomaker/Tmem8c) contribute to the formation of myofibers by promoting myoblast fusion via a mechanism that involves appropriate localization of Tmem8c at the plasma membrane of myoblasts allowing trafficking related to palmitoylation of C-terminal cysteine residues and C-terminal leucine [17]. Amino acid supplements (e.g., leucine, a master dietary regulator of muscle protein turnover, and its metabolite β-hydroxy β-methylbutyrate) and early refeeding with high protein diet (especially fast digestive proteins) can preserve muscle mass and function, revert sarcopenia, and enhance mobility and quality of life (QoL) by correcting age-related nutritional deficiencies, muscle protein turnover, and immune dysregulation—these effects are even greater when combined with other nutrients like vitamin D or omega 3 fatty acids as well as with physical exercise [6,131,132,133,134,135,136,137,138].

Research indicates that age-related skeletal muscle wasting results mainly from insufficient delivery of amino acids to skeletal muscle due to dysregulations in the activity of mTORC1- and activating transcription factor-4 (ATF4)-mediated amino acid sensing pathways. Meanwhile, interventions that ameliorate age-related damages in skeletal muscle operate primarily by reversing alterations in the delivery of amino acids to skeletal muscle via upregulation of mTORC1 and/or ATF4 [139]. mTORC1 is a nutrient sensing protein that acts as a core regulator of protein metabolism. It is sensitive to amino acids, energy status (ATP), stress (e.g., oxidative stress), and growth factors (e.g., insulin), which all can regulate its signaling [2,136]. Nevertheless, bioavailability of amino acids is necessary for growth factors to effectively activate mTORC1. Even more, amino acids on their own can adequately activate mTORC1 [136]. Evidence from preclinical and human studies confirms that ingestion of essential amino acids (similar to those found in royal jelly and bee pollen such as valine) increases cellular bioavailability of amino acids, which is associated with activation of the endothelial nitric oxide synthase (eNOs) pathway. eNOs further upregulates mTORC1 kinase. Translocation of mTORC1 from the cytosol to the surface of lysosomes is associated with improved mitochondrial biogenesis and cellular oxidative capacity in skeletal muscle due to activation of its substrates: P70 ribosomal proteins S6 kinase (S6K) and eukaryotic translation initiation 4E-binding protein 1 (eIF4E, also known as 4eBP1) [120,135,136].

As shown in Table 1, royal jelly and 10-HDA significantly increased muscle mass [54,96,100] and improved motor performance in aged rats [95,98,99]. In addition, dietary supplementation with monofloral bee pollen significantly improved the rate of muscle protein synthesis and restored muscle mass in emaciated old rats via upregulation of mTOR and two related downstream controllers of protein translation: p70S6k and 4eBP1, which were suppressed in malnourished old rats [36]. Although propolis improved various muscle-related parameters, its effect on muscle mass in rodents was limited—relative to royal jelly and bee pollen. However, it fostered muscle protein deposition in post-larva Nile tilapia [104]. Moreover, milk naturally enriched with PUFA and polyphenols from propolis remarkably increased the weight of the gastrocnemius muscle of growing obese rats while whole milk and milk enriched with PUFA only could not express any effect on skeletal muscle [105]. Therefore, this finding denotes that propolis could have enhanced the delivery of amino acids available in milk to skeletal muscle leading to its growth. Altogether, it is likely that the observed anabolic effects of royal jelly and bee pollen are associated with their high content of proteins and amino acids [32,36,83,84,85,86].

### 4.5. Suppression of Catabolic Activity in Skeletal Muscle

Skeletal muscle tissues represent the largest protein store in the human body (30–45% of total protein). Muscular mass, strength, and functions are greatly governed by the rates of muscle protein synthesis and turnover [124]. Muscle protein metabolism is regulated by the interaction of a wide range of genes. Aging is associated with various stresses, which increase the expression of catabolic genes such as E3 ubiquitin ligases MuRF1 and atrogin-1 (MAFbx). These genes heighten the occurrence of age-related muscular atrophy [97]. Oral consumption of royal jelly by aged HET mice resulted in lower levels of catabolic genes (e.g., MuRF1 and MAFbx), which were similar to those in young mice. In the meantime, the expression of these genes in the control mice was high indicating that royal jelly can delay age-related muscular apoptosis by suppressing the activity of catabolic genes [97]. Two other studies reported that CAPE suppressed catabolism that contributed to degenerative myopathy in the gastrocnemius muscle of rats undergoing eccentric exercising or femoral artery ligation as reflected by decreased serum levels of creatine kinase, a marker of muscular proteolysis [101,102,103]. Apart from skeletal muscle, CAPE was reported to protect heart muscle against age-related deteriorations such as accumulation of lipofuscin, nuclear irregularity, mitochondrial degeneration, and myofilament disorganization and disruption [33].

The molecular mechanism underling blockage of muscle proteolysis by bee products in sarcopenic rodents involves an interplay of various signaling pathways. Royal jelly and bee pollen activated mTOR and its substrate AKT, which are suggested to inhibit muscular proteolysis [36,96]. Similar to the effect of royal jelly on catabolic genes in HET mice, treating both L6 myoblasts and rats with propolis, CAPE, and kaempferide resulted in potent activation of AKT in a PI3K-dependent manner [111], in addition to phosphorylation of IR, PI3K, and AMPK [70]. AKT, a key substrate of mTORC2, is a conserved serine/threonine nutrient sensing protein kinase that belongs to the PI3k-related protein kinase family. Upon presence of growth factors, PI3k gets activated by IR substrate resulting in stimulation of a series of signaling cascades that involve activation of AKT, which leads to further activation of mTORC1. mTORC1 activates the phosphorylation of two main regulators of cap-dependent protein synthesis: S6K and eIF4E [2,140]. In addition, mTORC1 contributes to autophagy—a turnover process that involves clearance of dysfunctional organelles and long-lived protein aggregations with provision of energy and macromolecular precursors in return—by binding with AMPK resulting in phosphorylation of autophagy genes such as Unc51-like kinase 1 at different sites [140]. In fact, royal jelly is reported to fine-tune the transcriptional activity of the FOXO through modulating the activity of insulin/IGF-1 signaling [141]. FOXO plays a major role in the activation of AKT pathway, which implicates regulation of multiple stress–response pathways such as ROS detoxification and DNA repair and translation. In addition, the FOXOs family exerts a direct effect on certain muscle atrophy genes such as MUSA1 and a formerly uncharacterized ligase known as Specific of Muscle Atrophy and Regulated by Transcription (SMART) [142].

### 4.6. Enhancement of Stem Cell Function

Reduction of the number and functional capacity of the muscle satellite cells is considered a core contributor to the development of age-related muscular dysfunction [96]. Induction of myogenesis via in vivo reprogramming of muscle satellite cells is a currently studied strategy that has not been successfully used for sarcopenia treatment yet [143]. On the other hand, treating sarcopenic rats with both royal jelly and pRJ was reported to increase the number of Pax7-positive satellite cells in vivo and in vitro (pRJ only). pRJ induced self-renewal of satellite cells via activation of AKT signaling [96,97]. AKT activity was associated with activation of IGF-1 as indicated by increased serum levels of IGF-1. IGF-1 plays a crucial rule in the activation of various signaling pathways; it is believed to be a major mediator of muscle growth and repair that functions by stimulating the proliferation and differentiation of satellite cells into myotubes, albeit the exact mechanism is not clear yet [96]. Similarly, pRJ activated AKT-signaling pathway in satellite cells culture, which was associated with increased proliferation of myosatellite cells and their differentiation into myotubes—an effect that is contradictory to muscle loss. AKT is thought to contribute to the synthesis of muscular proteins and inhibition of muscle proteolysis [96].

### 4.7. Counteracting Glycation Stress

Oxidative stress, inflammation, and insensitivity to insulin, which accompany advanced age, contribute to the production of advanced glycation end products (AGEs) via enhancement of the activity of the Receptor for Advanced Glycation End products (RAGE) [144,145]. AGEs destroy the protein and lipid ingredients of muscle tissues by promoting the production of destructive molecules such as free radicals and inflammatory cytokines [101,102,113]. Thanks to their potent antioxidant properties, polyphenolic compounds exert multifaceted anti-glycation functions. On one hand, they scavenge free-radicals and chelate transition metals that are involved in the synthesis of dicarbonyl intermediates subsequently resulting in inhibition of the formation of AGEs. On the other hand, polyphenolic compounds antagonize AGE receptors, mainly RAGE, and facilitate the removal of already formed a,b-dicarbonyl intermediates such as methylglyoxal, promoting the degradation of AGEs [37,38,146]. Royal jelly is reported to downregulate the activity of RAGE, the main receptor for AGEs, in an aged model of cognitive impairment [147]. However, its anti-glycation effect has not been investigated in skeletal muscle yet. Propolis exhibits strong anti-AGE properties, which are superior to those of quercetin or chlorogenic acid, well-known natural AGE inhibitors. Its flavonoid fraction potently impedes the synthesis of AGEs by trapping dicarbonyl intermediates [37]. Table 1 shows that propolis accelerated AGEs clearance in a model of muscle aging induced by administration of a precursor of AGEs (methylglyoxal) via activation of glyoxalase 1, an enzyme that eliminates dicarbonyl compounds (key elements of AGEs) [38]. CAPE inhibited the production of AGEs-related molecules such as protein carbonyl in the gastrocnemius muscle of rats via blockage of the activity of xanthine oxidase and adenosine deaminase [101,102,103]. The latter negatively affects insulin signaling and promotes the development of hyperglycemia, which represents a favorable condition for the production of AGEs [148]. However, propolis could not counteract the wasting effects of AGEs that already occurred in the extensor digitorum longus muscle though it tended to restore soleus muscle mass. This finding denotes that different muscle tissues respond differently to treatment, probably based on their ratio of type I to type II fibers. It also signifies the importance of early use of bee products (e.g., propolis) for the prevention of AGEs formation in skeletal muscle in people with high risk for AGEs formation such as diabetics [38].

### 4.8. Neuronal Regeneration

Neuronal denervation is a key factor that contributes to skeletal muscle loss, and it is related to a plethora of pathological conditions [3,149]. Experimental induction of oxidative stress and inflammation results in skeletal muscle atrophy through induction of denervation e.g., of sciatic nerve [150]. Injuries of peripheral nerves (e.g., sciatic nerve) interrupt mechanical transmission and microvasculation of the nerve and induce reperfusion. Reperfusion involves pooling of oxygen and nutrients promoting high emission of free radicals, which attack protein and lipid contents surrounding the injury site resulting in excessive tissue loss [113]. Likewise, alterations in gut microbiome in aged rats are associated with alterations in serum level of vitamin B12 and fat metabolism as well as reductions in the gastrocnemius muscle mass and sciatic response amplitude [151]. Furthermore, dysregulation of insulin-mediated GLUT4 activity in certain areas of the central nervous system impairs neuronal metabolism and plasticity [69]. Meanwhile, activation of PGC-1α, a core regulator of mitochondrial content and oxidative metabolism, increases muscle fiber resistance to denervation and atrophy through downregulation of two ubiquitin-ligases involved in the ubiquitin-proteasome pathway: MuRF1 and Atrogin-1 [1,152].

Propolis treatment for four weeks restored gastrocnemius muscle weight and improved functional performance (e.g., walking) in rats with crush injury of the sciatic nerve. Effects of propolis were associated with increased nerve healing and regeneration as depicted by faster healing of the myelin sheath and ultra-structurally normal unmyelinated axons and Schwann cells. Investigations of motor conduction from the sciatic nerve to the gastrocnemius muscle indicated that nerve recovery induced by propolis treatment promoted optimal physical functioning by allowing motor conduction to reach the gastrocnemius muscle [113]. Neuroprotective effects of propolis in motor neurons are documented in the literature. Both kaempferide and kaempferol protected motor neurons against atrophy induced by the toxic copper-zinc superoxide dismutase in amyotrophic lateral sclerosis—a serious neurodegenerative disease that involves selective and progressive loss of motor neurons [65]. In addition, orally administered chrysin (a flavonoid that is copious in propolis) to rats intoxicated by 6-hydroxidopamine showed neuroprotective effects by mitigating neuroinflammation, enhancing levels of neurotrophins and neuronal recovery factors (e.g., brain derived neurotrophic factor and glial cell line-derived neurotrophic factor), and maintaining integrity of dopaminergic neurons resulting in better motor performance [72].

The release of acetylcholine (a neurotransmitter that regulates cognition) at the synaptic cleft of the neuromuscular junction is essential for motor neurotransmission, which controls excitation-contraction coupling and cell size. However, free radicals, cytokines, and AGEs impair neurotransmission by altering the production of acetylcholine [6,21,149,153]. On the other hand, upregulation of acetylcholine receptors improves neurotransmission [154]. Treatment with royal jelly may correct acetylcholine neurotransmission given its high content of acetylcholine (4–8 mM) [60]. In addition, royal jelly, propolis, and bee pollen are rich in antioxidant elements that have a potential to scavenge ROS and mitigate other pathologies that contribute to acetylcholine deficiency (e.g., neuroinflammation) [73,101,102,103]. In this respect, treatment of experimental models of carrageenan-induced hind paw edema with hydroalcoholic extract of red propolis and its biomarker, formononetin, is reported to inhibit leukocyte migration and ameliorate inflammatory neurogenic pain induced by injections of formalin and glutamate [115]. However, investigations of the action of bee products on neurotransmission are very scarce.

### 4.9. Improving Muscular Blood Supply

Aging is associated with higher onset of atherosclerosis and restenosis, which involve vascular and microvascular damages that result from hyperproliferation of vascular smooth muscle cells (VSMCs) [155]. Muscle unloading (e.g., in sedentary lifestyle) involves chronic neuromuscular inactivity, which results in reductions in capillary number, luminal diameter, and capillary volume as well as heightened production of anti-angiogenic factors, such as p53 and TSP-1 in skeletal muscle [110]. Microvascular alterations and impaired nitric oxide (NO) production are key causes of decreased blood flow to the skeletal muscle. Poor muscular blood supply induces muscle wasting via a mechanism that entails impaired glucose metabolism and suboptimal protein anabolism [156,157]. In addition, ischemic injury in skeletal muscle is associated with high ROS release from polymorphonuclear leukocytes, which infiltrate muscle tissues. Free radicals alter cellular structure and function by attacking lipid and protein biomolecules that exist in the structure of biological membranes, enzymes, and transport proteins [102]. Therefore, improving vascularization and blood supply to skeletal muscle is a possible mechanism for the prevention of muscle wasting. 

Mitchell and colleagues examined changes in microvascular blood volume, microvascular flow velocity, and microvascular blood flow in the quadriceps muscle following treatment with 15 g of amino acids in young and old subjects (20–70 years old). They detected improvements in all the 3 parameters only in young groups, and those effects were associated with proper insulin activity. Thus, the authors suggested that refeeding effects on muscular blood supply may be hindered by dysfunctions in postprandial circulation and glucose dysregulation [157]. Bee products such as bee pollen and propolis have a potential to boost microcirculation and correct pathologies that contribute to vascular dysfunction such as hyperglycemia and dyslipidemia [36,155]. In this context, CAPE, one of the basic constituents of propolis, was reported to combat vascular damages by counteracting the proliferative activity of platelet-derived growth factor. The molecular mechanism involved inducing cell-cycle arrest in VSMCs via activation of p38mitogen-activated protein kinase (MAPK), which was associated with accumulation of hypoxia-inducible factor (HIF)-1α—mainly due to inhibition of HIF prolylhydroxylase, a key contributor to proteasomal degradation of HIF-1α—and subsequent induction of HO-1 [155]. In the same regard, supplementing rats undergoing capillary regression (resulting from two weeks of hind limb unloading) in the soleus muscle with two daily intragastric doses of propolis for two weeks restored capillary number, capillary structure, capillary volume, and capillary to muscle fiber ratio through a mechanism that involves inhibition of anti-angiogenic factors and activation of pro-angiogenic factors (e.g., VEGF). The relative muscle-to-body weight in treated animals was higher than in the unloaded control animals, and the number of TUNEL-positive apoptotic endothelial cells in the soleus muscle was similar to that in the normal control animals [110]. Likewise, bee pollen (both neat and processed) increased blood vessels in the gastrocnemius muscle of rats undergoing muscle injury because of vigorous exercise [90]. Treating C2C12 myoblasts with propolis ethanolic extract increased VEGF production [108]. VEGF is a pro-angiogenic factor that contributes to angiogenesis by recruiting endothelial cells and promoting their differentiation to form new vascular networks. VEGF protects skeletal muscle undergoing unloading against the progression of capillary regression [110]. Its expression in skeletal muscle and associated angiogenic response increase following exercise due to change in the activity of HIF-1α and AMPK. It promotes the generation of ATP following mitochondrial biogenesis in order to meet oxygen demand [158]. Altogether, these reports signify that bee products and endurance exercise share common mechanisms to produce their vitality effects in skeletal muscle.

### 4.10. Improving the Composition of Gut Microbiome

Bee products such as royal jelly and propolis display potent antifungal, bactericidal, microbicidal, anti-inflammatory, antioxidant, and healing effects [2,66]. Most bee products investigated in this review were administered orally. Therefore, it is likely that their therapeutic effects may start locally within the GI tract, which frequently undergoes propagation of harmful endobacteria, inflammation, aberrations, and permeability in advanced age [10,24,152]. In this respect, Roquetto and colleagues [116] supplemented C57BL/6 mice on HFD with crude propolis (0.2%) for two and five weeks. HFD increased the proportion of the phylum Firmicutes as well as levels of circulating lipopolysaccharide (LPS) and inflammatory biomarkers. DNA sequencing for the 16S rRNA of the gut microbiota revealed that five weeks of propolis treatment rendered the microbiota profile almost normal. Compared with untreated mice, propolis-supplemented animals demonstrated lower levels of serum triacylglycerols, glucose, and circulating LPS, along with reduced expression of TLR4 and inflammatory cytokines in skeletal muscle [116].

Lactic acid bacteria profusely exist in bee saliva and all bee products [36,83,84]. Various species of lactic acid bacteria have been experimentally used to correct GI dysbiosis and related muscle wasting [159,160]. Moreover, oligosaccharides have been chemically isolated from bee products [32,161]. These api-materials are classified as prebiotics, fermented non-digestible compounds that promote the proliferative activity of health-promoting bacteria [24,161]. Supplementing frail old adults with fructooligosaccharides expressed positive effects on skeletal muscle strength (handgrip) and endurance (exhaustion) [162]. On the other side, microbiome of the gut can affect the biological activity of bee products. The literature shows that certain endobacteria transform dietary polyphenols into phenolic acids, which can easily access the circulation and then cross the blood brain barrier to produce therapeutic effects [163]. Hence, it is important that future investigations of bee products among sarcopenic subjects examine the effect of these products on the composition of GI microbial population and its association with muscle-related outcomes.

## 5. Discussion

Age-related deteriorations in skeletal muscle are multifactorial in nature, which contributes to the high failure rates of drugs designed to target sarcopenia. Thus, there is a strong need for new therapeutic agents that express multidimensional effects in order to provide physically frail elders a chance to restore their health and QoL [21]. Given the scarcity of human trials using bee products among physically frail and sarcopenic subjects, cell culture and animal studies may provide useful information about the effect of bee products on skeletal muscle and help identifying their probable mechanism of action. Both in vivo and in vitro studies indicate that substances produced by honey bees such as royal jelly, bee pollen, and propolis can positively modulate basic cellular functions of myoblasts as well as mature myocytes. These products seem to be promising anti-sarcopenic agents because they can promote muscle protein synthesis [36,104], decrease mitochondrial oxidative stress [36,90,106], mitigate inflammation [38,90,101,108,115], improve muscular blood supply [102,110], enhance peripheral motor conduction [113], accelerate the removal of AGEs [38], counteract catabolism, and decrease markers of skeletal muscle atrophy [97,101,102,103].

One of the major pathologies underlying age-related skeletal muscle failure is insulin resistance, which alters glucose metabolism [3,4,21,118,130,164]. Numerous studies signify that bee products may protect against muscle wasting by correcting metabolic dysregulations in skeletal muscle via activation of some of the key nutrient sensing molecules [69,96,100,106,109,111]. In this respect, bee pollen permitted mTORC1 to mitigate anabolic resistance through the regulation of muscle protein and muscle growth in malnourished sarcopenic old rats [36]. Royal jelly and CAPE restored muscle mass and metabolism through activation of AKT, a substrate of mTOR [96,111]. Polyphenols in propolis [70], CAPE [111], and 10-HDA [106] activated AMPK resulting in enhanced skeletal muscle glucose uptake. In the meantime, royal jelly improved mitochondrial biogenesis in skeletal muscle via activation of AMPK [106]. AMPK and mTOR are master regulators of metabolism and autophagy [2,165]; they keep muscle integrity by stimulating protein synthesis [21,22]. Accordingly, all these findings denote that bee products represent a promising nutritional strategy that can tackle skeletal muscle wasting in old age by correcting its underling metabolic causes. However, the detailed physiological processes involved still need to be deeply unraveled in further investigations.

The activity of mTOR depends mainly on the bioavailability of essential amino acids in addition to many other molecules in the cellular microenvironment [136]. Evidence from preclinical and human studies confirms that ingestion of essential amino acids increases their cellular bioavailability, which is associated with upregulation of the activity of the nutrient sensing mTORC1 kinase following its translocation from the cytosol to the surface of lysosomes [135,136]. Research shows that ingestion of small amounts of essential amino acids (6.8 g) by healthy older adults could stimulate muscle protein synthesis by activating mTORC1 signaling even without involvement in resistance exercise training [137]. Other factors affect the capacity of essential amino acids to stimulate muscle protein synthesis. Amino acid bioavailability depends to a great extent on the bacterial strains of resident gut microbiome, which can either promote amino acid loss by degrading them or substantially contribute to their availability by producing some of them e.g., lysine [152]. Furthermore, amino acid supplements in old age do not enhance muscle protein metabolism, combat anabolic resistance, or improve muscle condition under sedentary conditions [25,166]. 

According to Table 1, different models of skeletal muscle injury (e.g., natural aging [96], sarcopenic obesity [54], AGE-induced muscle wasting [38], malnutrition-related muscle loss [36], exhaustive exercise [90,109], etc.) were used to evaluate the effect of bee products on skeletal muscle. The effects of bee products on muscle mass varied considerably: some studies reported that bee products increased muscle mass [36,54,90,96,100,105,113] and improved physical performance [95,96,97,98,99] while others could not or did not depict any significant change in muscle mass [38,102,103,104,110,117]. However, the latter revealed major beneficial effects related to the biology of skeletal muscle aging such as improved muscle protein deposition [104], enhanced activity of mitochondrial enzymes [106], decreased muscle infiltration by inflammatory cells, decreased muscle proteolysis, lowered lipid peroxidation and protein carbonylation [102,103], increased microvascular blood supply, heightened production of antioxidants [110], and increased clearance of AGEs from skeletal muscle [38]. These findings suggest that bee products may prevent the development of sarcopenia if supplemented earlier before the occurrence of muscle atrophy. In this regard, supplementing young football players with royal jelly for two months resulted in a significant increase in muscle and bone mass compared with control players who did not receive royal jelly [167].

As for the effect of bee products on sarcopenia in humans, we could locate only one randomized control trial (RCT), which examined the effect of pRJ (1.2 g/d or 4.8 g/d over 1 year) on muscle strength among institutionalized aged population. This study revealed no significant effect of pRJ on the tested muscular functions: handgrip strength, six-minute walk test, timed up and go test, and standing on one leg with eyes closed [59]. Nonetheless, such outcome indexes may not be sufficient to evaluate the overall effect of royal jelly on muscular performance and quality. In addition, the literature documents gut microbiome alterations-related to muscle loss in institutionalized old adults compared with older adults living in the community, which might affect the efficacy of nutritional therapies in this particular group [164,168,169]. 

The relation between muscle mass or strength and physical function in old people is not as direct or clear as originally presumed. Existing knowledge indicates that muscle strength may be intact in people with muscle loss. Likewise, decreased muscle strength may not necessarily alter physical performance [21,170]. Research notes that selection of interventions that merely increase muscle mass and/or strength entails ignoring valuable therapies that contribute to various biological processes of muscle recovery by interfering with pathologies conducive to muscle loss. More, combining functional measures of muscle performance with different comprehensive biomarkers (e.g., of metabolism, inflammation, oxidative stress, etc.) in skeletal muscle and the whole body may identify therapies that can positively affect muscle qualities independent of mass increase [21]. In support of this argument, treating sarcopenic elderly with oral amino acids (which represent only one ingredient of royal jelly or bee pollen) significantly increased whole-body lean mass 6 and 18 months after treatment as indicated by dual-energy X-ray absorptiometry. This effect was attributed to enhancement of insulin sensitivity and anabolism as portrayed by significant reductions in fasting blood glucose, serum insulin, HOMA-IR, and serum tumor necrosis factor-α (TNF-α) as well as increase of serum level of IGF-1 and IGF-1/TNF-α ratio [171]. Animal studies lend further support to those reports. For instance, royal jelly had no effect on muscle differentiation genes (myogenic differentiation 1 (MyoD), myogenein, myostatin) in sarcopenic mice, indicating that royal jelly may not improve age-related deterioration of muscle strength—once it occurs. On the other hand, royal jelly significantly reduced the progression of muscle atrophy by decreasing the expression of catabolism genes E3 ubiquitin ligases MuRF1 and atrogin-1 in old mice to levels similar to those in young mice [97]. Royal jelly also stimulated the differentiation of satellite stem cells both in vivo and in vitro and improved the regenerative capacity of injured muscle [96,97]. Therefore, royal jelly treatment might reduce the progression of age-related muscular atrophy [59]. Future RCTs that evaluate the effect of bee products on muscle qualities and motor performance in humans should be properly designed to include biological markers of motor functioning along with behavioral measures.

Multifactorial conditions such as sarcopenia should be addressed by multimodal interventions. Nutrition and physical exercise are key strategies that can preserve muscle mass and function in older adults both in clinical and community settings [21]. Thus, it might be helpful to compare the effect of combining bee products with other conventional treatments such as other dietary elements and exercise given that factors such as physical activity and diet can interfere with or promote the effect of bee products on muscle qualities and disorders that underlie muscle weakness [105,106]. Fat mass increases with age, especially after age-related hormonal decline (e.g., after menopause), and it activates inflammatory processes that disturb body physiology [172]. In this regard, supplementing obese rats with milk naturally enriched with PUFA and polyphenols from propolis increased gastrocnemius muscle mass and tended to increase the mass of the soleus muscle. It decreased the diameter of adipocytes and tended to decrease serum levels of low-density lipoprotein. These findings suggest a role of propolis-enriched milk in the mitigation of sarcopenic obesity, albeit it had no effect on body weight [105]. On the other side, many studies show that exercise has significantly improved muscle mass and strength in sarcopenic seniors [4,42,132,173]. Propolis treatment of undifferentiated L6 myoblast selectively stimulated IL-6 production and inhibited pathological cytokines such as interleukin (IL)-1β and TNF-α [108]. This interesting in vitro investigation indicates that propolis can mimic the mechanism through which exercise induces skeletal muscle remodeling [23,114]. Evidence from preclinical studies shows that concurrent treatment of rats on endurance training with royal jelly enhanced mitochondrial adaptation in muscles that combine both type I and type II fibers such as the soleus muscle whereas neither royal jelly alone nor exercise alone could influence the activity of mitochondrial enzymes in that muscle [106]. Moreover, bee pollen and propolis treatment of rats on exhaustive exercise increased the production of antioxidant enzymes, blocked the production of free radicals, promoted glycogen use in the skeletal muscle and liver, and restored muscle fiber structure [90,109]. 

In most studies, whole royal jelly, bee pollen, and propolis were used. However, these products appeared in different forms e.g., neat vs processed bee pollen [90], water [109] vs ethanolic extracts [110] of propolis, lyophilized [98,99] and crude [97] vs enzyme-treated [59,95,96,97] royal jelly. Both processed bee pollen [36] and pRJ [97,106] had better effects compared with crude bee pollen and royal jelly. In addition, a large group of constituents of royal jelly and propolis were used such as 10-HDA [54,106], CAPE [69,101,102,103], artepillin C, coumaric acid, kaempferide [70], boropinic acid, 4-geranyloxyferulic acid, 7-isopentenyloxycoumarin, and auraptene [69]. 10-HDA was the only element in royal jelly that was tested in skeletal muscle. It enhanced glucose uptake via AMPK phosphorylation [106]. It also restored body weight, restored skeletal muscle mass (only in males), and reduced fat mass (only in females) in aged rats undergoing chronic stress [54]. CAPE is one of the most investigated compounds in propolis: it enhanced skeletal muscle glucose uptake [111], inhibited cytokine and ROS production, prevented protein carbonylation, lipid peroxidation, and muscle proteolysis [101,102,103]. The effects of whole ethanolic extracts of propolis and CAPE on glucose uptake in skeletal muscle were comparable to those of insulin [69,109,111]. More, investigations of the effect of flavonoids and oxyprenylated phenylpropanoids abundant in ethanolic extracts of propolis on glucose uptake in skeletal muscle revealed superior effects of kaempferide [70], 4-geranyloxyferulic acid, and auraptene. Among 5 oxyprenylated phenylpropanoids derived from propolis, auraptene most potently activated GLUT4 translocation and accelerated glucose influx into skeletal muscle cells. Measurement of the incorporated amounts of these compounds into myotubes indicated that auraptene had the highest bioavailability among other effective compounds [69]. 

Matters concerning dosage and duration of treatment remain issues of concern should these compounds be used to prevent and treat sarcopenia. In addition, the effect of other active ingredients of bee products on skeletal muscle pathologies is worth investigation. For instance, MRJPs are reported to contribute to most therapeutic properties of royal jelly due to their rich amino acid content (up to 578 amino acids) [47]. In the meantime, the literature documents a potent regulatory effect of amino acids on skeletal muscle protein turnover [6,131,132]. Nonetheless, the effect of MRJPs on skeletal muscle were not studied until now. 

Despite the details illustrated in this review, many questions remain unanswered. The most important question among all is which bee product or bee component can produce the best benefits against skeletal muscle senescence and under which circumstances? Sarcopenia is the result of several interrelated factors such as neuromuscular dysfunction, oxidative stress, metabolic alterations, poor blood supply, hormonal deficiencies (e.g., sex hormones), chronic inflammation, and lifestyle choices (e.g., physical inactivity and unhealthy diet) [21]. The effect of bee products on some of these factors are underaddressed (e.g., neuromuscular dysfunction, muscle blood supply, and gut microbiome) while some other have not been explored yet (e.g., sex steroids and their association with muscular function). 

## 6. Conclusions

All animal studies discussed above indicate that royal jelly, propolis, and bee pollen as well as their key ingredients such as 10-HDA and CAPE might counteract age-related muscular decline, especially in early stages. These products operate by modulating most mechanisms that contribute to sarcopenia such as metabolic dysregulation, inflammation, oxidative stress, etc. However, more studies are needed to examine the specific cellular and molecular mechanisms of these products and other major ingredients that were not explored, such as MRJPs in royal jelly. Among all bee products, royal jelly has been used to improve muscular performance in humans, though at a small scale, without any reported side effects. Future RCTs that examine the effect of bee products on sarcopenia should consider individual variations (e.g., gender, general health, activity level, diet, etc.) and combine functional and subjective outcome measures with sound predictive biomarkers.

## Figures and Tables

**Figure 1 foods-09-01362-f001:**
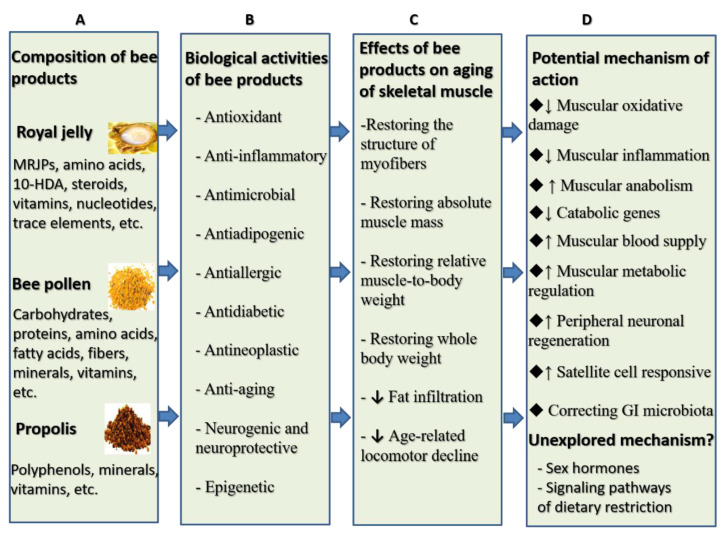
Brief summary of the chemical composition and biological activities of royal jelly, bee pollen, and propolis along with their effects on aged skeletal muscle and possible underlying mechanism. ↑ denotes increase, ↓ denotes decrease, MRJPs: major royal jelly proteins, 10-HDA: trans-10-hydroxy-2-decenoic acid, GI: gastrointestinal. Panels (**A**,**B**) describe the composition and the biological properties of royal jelly, bee pollen, and propolis. Thanks to their rich structure (which consists of proteins, polyphenols, vitamins, minerals, and trace elements) [2,28,37,49,59,60,61,68,80,86], these bee substances demonstrate a wide range of therapeutic activities such as counteracting inflammation and oxidative stress, in addition to many others [37,38,39,43,45,63,83]. Panel (**C**) summarizes the effects of royal jelly, bee pollen, and propolis on skeletal muscle. These products increased muscle mass [36,54,95,96] and restored muscle function in old [36,38,95,96,97] and exhausted [54,90,101,106] animal models. As shown in Panel (**D**), these effects originated from multiple molecular events that resulted in several therapeutic actions including amelioration of inflammation [36,38,90,101,107,108] and oxidative [95,106,109,110] damages, metabolic regulation [69,70,96,100,106,109,111,112], enhancements of satellite cell responsiveness [96,97], improving muscular blood supply [110], inhibition of catabolic genes [97], and promotion of peripheral neuronal regeneration [113]. However, future studies could examine the involvement of other possible mechanisms in the muscle-enhancing potential of these bee products such as the role of gut microbiome in the absorption of nutrient contents of bee products. It is not clear if these api-materials affect muscle quality via modulation of sex steroids and signaling pathways of dietary restriction, which are known to affect muscle integrity.

**Table 1 foods-09-01362-t001:** Characteristics of Included Pre-Clinical and Clinical Studies Examining the Effect of Royal Jelly, Propolis, Bee Pollen, and their Constituents on Skeletal Muscle (N of studies = 24).

Bee Products/Their Constituents	Animal Model/Cell Line	Experimental Design	Outcome Measures	Results and Possible Related Mechanisms	References
**In vitro** Royal jelly and pRJ (500 μg/mL: 1, 2, 3, and 5 d) **In vivo** Dietary royal jelly and pRJ (1% and 5% for 3 months)	**In vitro** SC isolated from aged mice **In vivo** 21-months old male C57BL/6 mice as a mouse model of sarcopenia	**In vitro** EG1: royal jelly treated SC EG2: pRJ treated SC CG: untreated SC **In vivo** EG1 and EG2: aged mice on 1% and 5% royal jelly, respectively EG3 and EG4: aged mice on 1% and 5% pRJ, respectively CG: untreated aged mice	**In vitro** SC proliferation and differentiation into myotubes and AKT signaling. **In vivo** No of SC, skeletal muscle weight, grip strength, regenerative capacity of injured muscle, and serum IGF-1.	**In vitro** pRJ enhanced SC proliferation rate and differentiation into myotubes through activation of AKT signaling. **In vivo** Royal jelly and pRJ significantly increased the number of SC, weight of skeletal muscle, grip strength, regenerative capacity of injured skeletal muscle, and IGF-1 serum level.	[96]
Intragastric 10-HDA (1.6 mM/kg body weight)	**In vitro** L6 myotubes obtained from Osaka biobank **In vivo** 7-weeks old male C57BL/6J mice	**In vitro** EG: 10-HDA CG1: AICAR (1 mM) CG2: DMSO (0.1%) **In vivo** EG: 10-HDA (1.6 mM) CG1: gum Arabic (5%)	Glucose uptake, AMPK signaling, and Glut4 translocation.	10-HDA increased glucose uptake into L6 myotubes following AMPK activation and Glut4 translocation to the plasma membrane. AMPK activation was induced by the upstream kinase Ca²/calmodulin-dependent kinase β, independent of changes in AMP:ATP ratio and the liver kinase B1 pathway.	[106]
Intragastric royal jelly and pRJ (0.7 and 1.4 mg/kg body weight/d/90 d)	D-galactose induced mouse model of aging	EG1 and EG2: mice on 0.7 and 1.4 mg RJ, respectively EG3 and EG4: mice on 0.7 and 1.4 mg RJ hydrolysate, respectively CG: untreated mice	Antioxidant enzymes, body weight, muscular performance, memory, and learning.	Both doses of pRJ prevented age-related locomotor decline, preserved body weight, enhanced memory and learning, increased antioxidant enzyme activity, and inhibited the production of lipid peroxides.	[95]
Oral 10-HDA (12-60 mg/kg body weight/d/4 months)	Obese old rats and stressed mice as models of sarcopenia and depression (male and female Sprague-Dawley and 2-months old male and female BALB/c)	EG1 and EG2: aged obese rats on 10-HDA (12 or 24 mg/kg/d for 3.5 months) EG3 and EG4: stressed BALB/c mice on 10-HDA (30 or 60 mg/kg/d for 4 months) CG1 and CG2: untreated aged obese rats and untreated stressed BALB/c mice	Body weight, weight of abdominal adipose tissue, and muscle mass.	10-HDA significantly increased weight gain and weight maintenance in aged rats/mice undergoing behavioral stress without any change in diet consumption. It also significantly decreased adipose tissue in female animals and increased muscle mass in male rodents compared with untreated controls.	[54]
Dietary royal jelly and pRJ + MF powder diet (0.05% or 0.5%)	6-months old male HET mouse model of severe sarcopenia (background strains: BALB/c, C57BL/6, C3H, and DBA/2)	EG1 and EG2: HET mice on 0.05% and 0.5% royal jelly, respectively EG3 and EG4: HET mice on 0.05% and 0.5% pRJ, respectively CG1: untreated HET mice CG2: untreated young mice	No of blood cells and Pax7 SC, albumin, AST, ALT, T-CHO, TG, expression of muscle genes (MyoD, myogenein, myostatin) and catabolic genes (E3 ubiquitin ligases MuRF1, and atrogin-1). Behavioral tests: grip strength, wire hang, rotarod, and horizontal bar tests.	RJ and pRJ significantly delayed age-related impairment of motor functions, positively improved physical performance of treated mice in 4 types of tests (grip strength, wire hang, horizontal bar, and rotarod), lowered age-related muscular atrophy, increased No of Pax7 SC markers, and suppressed catabolic genes.	[97]
Intragastric royal jelly (100 mg/kg body weight/d/8 weeks)	10-months old male Sprague-Dawley on HFD as a rat model of sarcopenic obesity	EG1: aged rats on royal jelly and HFD EG2: aged rats on royal jelly CG1: untreated young rats CG2: untreated aged rats CG3: untreated old rats on HFD	Serum levels of T-CHO, TG, HDL-c, LDL-c, insulin, HOMA-IR. Skeletal muscle TG levels. Serum and adipose tissue levels of TNFR1. Percentage of weight gain of the body, abdominal fat, and tibialis anterior and hind limb muscles.	Royal jelly significantly decreased insulin levels, HOMA-IR, TNFR1 in serum and adipose tissue, serum lipids, muscle TG levels, body weight gain and abdominal fat weight and significantly increased the weight of hind limb muscle in aged rats on HFD compared with aged mice on HFD only.	[100]
Oral royal jelly (1.0 mg/g body weight/d/3 weeks)	6–7 weeks old ICR mice	EG: endurance exercise + royal jelly CG1: sedentary rats on royal jelly CG2: endurance exercise+ distilled water CG3: sedentary rats on distilled water	CS, β-HAD, and activities of AMPK and acetyl-CoA carboxylase in the soleus, plantaris, and tibialis anterior muscles.	Royal jelly increased CS and β-HAD maximal activities in the soleus muscle compared with all CGs but failed to affect these enzymes in the plantaris and tibialis anterior muscles of sedentary mice compared with CG2. Royal jelly effects in the soleus muscle were mediated by AMPK and acetyl-CoA carboxylase activity.	[106]
Gavage/oral lyophilized royal jelly (50 and 100 mg/kg body weight/d/8 weeks)	18-months old (naturally aging) Wistar male rats as a model of aging	EG1 and EG2: aged rats on royal jelly 50 and 100 mg, respectively CG: aged rats on gavage solution of 0.9% NaCl.	Learning, spatial memory, and motor performance on Morris water maze.	Royal jelly improved learning, spatial memory, and motor performance e.g., increased the number of crossings, swimming speed, and swimming distance.	[98,99]
Oral pRJ (1.2 or 4.8 g/d over 1 year)	Institutionalized older adults (mean age: 78.5 ± 7.5 years, N = 199, N males = 99, N females = 95)	EG1 and EG2: pRJ (1.2 and 4.8 g/d), respectively CG: placebo	Handgrip strength, six-minute walk test, timed up and go test, and standing on one leg with eyes closed.	pRJ had no significant effect on handgrip strength, six-minute walk test, timed up and go test, and standing on one leg with eyes closed.	[59]
Brazilian propolis extract (100 μg/mL/4-12 h)	**In vitro** Differentiated myoblast C2C12 cells and RAW264 macrophages isolated from mice	EGs: propolis (100 μg/mL) CG1: ethanol (0.008%) CG2: DMSO (0.08%). CG3: IKK inhibitor (BMS-345541)	IL-6, LIF, CCL-2, CCL-5, CXCL-10, VEGF-A, COX2, MMP-12, migration of RAW264 macrophages, and activities of MAIL/IκBζ and NF-κB.	Propolis (at 8h) induced RAW264 macrophage migration, activated MAIL/IκBζ and NF-κB proteins p50 and p65, and increased levels of VEGF-A, COX-2, MMP-12, CCL-2, CCL-5, CCL-10, LIF, and IL-6. Propolis inhibited the production of IL-1β and TNF-α.	[108]
CAPE (1 and 10 μM/3 min-12 h)	**In vitro** Differentiated L6 myoblast cells isolated from rats	EGs: CAPE (1, 10 μM) CG1: insulin (100 nM) CG2: AICAR	2-Deoxyglucose uptake, AMPK and AKT signaling.	CAPE (10 μM/1h) increased 2-Deoxyglucose uptake (same as insulin) and activated AMPK (same as AICAR, an AMPK activator). CAPE (10 μM/3 min) activated AKT in a PI3K-dependent manner.	[111]
Boropinic acid, 4-geranyloxyferulic acid, 7-isopentenyloxycoumarin, auraptene (0.1, 1, 10 μM), and raw Italian propolis (0.001–1 mg/mL)	**In vitro** Differentiated L6 myoblast cells isolated from rats	EGs: boropinic acid, 4-geranyloxyferulic acid, 7-isopentenyloxycoumarin, auraptene (0.1, 1, 10 μM), and propolis (0.001–1 mg/mL) CG1: insulin (0.1 μM) CG2: DMSO	GLUT4-mediated glucose uptake and GLUT4 translocation.	Propolis (1.0 and 1 mg/mL), 4-geranyloxyferulic acid, 7-isopentenyloxycoumarin, and auraptene significantly increased glucose uptake and GLUT4 translocation.	[69]
**In vitro** A single oral dose of Brazilian propolis extract (250 mg/kg body weight) **In vivo** Artepillin C, coumaric acid, and kaempferide (1–10^4^ ng/mL for 15 min)	**In vitro** Differentiated L6 myoblast cells isolated from rats **In vivo** 5-weeks old male ICR mice	EGs: artepillin C, coumaric acid, and kaempferide (1, 10 μM), and propolis (1–10^4^ ng/mL) CG1: insulin (100 nM) CG2: AICAR CG3: DMSO	2-Deoxyglucose uptake, OGTT, maltase and sucrase-isomaltase activities in epithelial cells of the small intestinal, phosphorylation of AMPK, PI3K, AKT, AS160, IR, and GLUT4 translocation.	Polyphenols in propolis activated PI3K and AMPK signaling pathways and promoted GLUT4 translocation in L6 myotubes though only kaempferide increased glucose uptake. Propolis extract (**In vitro**, 1 μg) and **In vivo** significantly promoted the phosphorylation of IR, PI3K, and AMPK and increased GLUT4 translocation in rat skeletal muscle and subsequently decreased postprandial blood glucose levels. Propolis extract had no effect on α-glucosidase activity in the small intestine.	[70]
Gavage CAPE (5 and 10 mg/kg/d/5 d)	6-7-weeks old male adult Wistar rats	EG: CAPE + eccentric exercise CG1: normal rats + propylene glycol in saline CG2: acute eccentric treadmill exercise	Serum creatine kinase levels, IL-1β, MCP-1, COX-2, iNOS, leukocyte infiltration, and the extent of muscle fiber damage (vacuolization and fragmentation).	CAPE decreased serum creatine kinase, protein nitrotyrosine, PARP activity, MDA, leukocyte infiltration, skeletal muscle cell fragmentation and vacuolization, muscle levels of COX2, iNOS, IL-1β, MCP-1, and p65NF-κB activity to levels in resting CG1 compared with CG2.	[101]
Dietary propolis (0.1% over 20 weeks)	MGO-induced muscle wasting in male C57BL/6NCr mice (4-weeks old)	EG: propolis + MGO CG1: MGO only CG2: propolis only CG3: untreated mice	Weight of EDL and soleus muscles, soleus and EDL levels of AGEs, inflammation-related molecules, and activity of glyoxalase 1.	Propolis had no effect on MGO-induced loss of EDL muscle but tended to increase the weight of the soleus muscle regardless of MGO treatment. Propolis decreased muscular levels of AGEs, IL-1β, IL-6, TLR4 and enhanced the activity of glyoxalase 1.	[38]
Dietary crude propolis (0.2% over 2 or 5 weeks)	HFD-induced muscle wasting in male C57BL/6 mice (4-weeks old)	EG: propolis + HFD CG1: HFD only CG2: untreated mice	16S rRNA of gut microbiota, serum levels of LPS, triacylglycerols and glucose, and skeletal muscle levels of inflammatory cytokine TLR4 expression.	Propolis (5 weeks) significantly decreased serum triacylglycerols, glucose, circulating LPS and down-regulated the expression TLR4 and inflammatory cytokine in muscle. It countereacted the effect of HFD on gut microbiota.	[116]
Oral propolis water extract (50 mg/kg body weight/d/6 weeks)	6-weeks old Sprague-Dawley rats	EG: propolis + eccentric exercise CG1: only eccentric exercise CG2: no treatment	Blood levels of glucose and insulin, MDA, SOD, GPX, and CAT in the liver and in the tissue of the liver and the gastrocnemius muscle.	Serum levels of glucose and insulin were significantly lower in EG and CG1 than CG2. Glycogen level in skeletal muscle was higher in EG and CG1 than CG2. Skeletal muscle levels of MDA were lower in EG than CG1 and CG2. Liver levels of SOD as well as gastrocnemius muscle levels of SOD, GPX and CAT were higher in EG only.	[109]
Gavage naturally-enriched milk with PUFA and propolis polyphenols (PUFA/P-M: 5 mL/kg body weight /85 d)	21-d old male Wistar rats	EG: HFD + PUFA/P-M CG1: HFD + water CG2: HFD + whole milk CG3: HFD + PUFA milk CG 4: standard chow + water N.B. All treatments were repeated in absence of HFD	Weight gain, mass of internal organs and the soleus and gastrocnemius muscles, and glucose tolerance.	Among all treatments in obese rats, only PUFA/P-m increased gastrocnemius muscle mass (tended to increase soleus muscle mass) and mesenteric fat and tended to lower LDL levels. It decreased the size of adipocytes compared with all groups except PUFA milk with no effect on body weight.	[105]
Dietary propolis 4% (105 d)	Nile tilapia in net cages (males only)	EG: propolis rich diet CG: propolis free diet	Muscle morphometry and myostatin gene expression.	Propolis diet had no effect on muscle growth or myostatin gene expression at 35, 70, and 105 d.	[117]
Dietary propolis (1, 2, 3 and 4 g/kg of feed/45 d)	Nile tilapia post-larvae and fingerlings in tanks	EG: propolis rich diet CG: propolis free diet	Final weight, total and standard body length, survival, body composition, and intestinal villus height.	Propolis supplementation had no effect on weight, total and standard length, survival, and intestinal villus height. However, 2.6 g propolis/kg of feed significantly improved body protein deposition and body condition factor—an estimate of future growth.	[104]
Intraperitoneal CAPE (10 μM/kg 1 h before ischemia reperfusion)	Adult male Wistar rats undergoing ischemia reperfusion	EG: ischemia reperfusion + CAPE CG1: ischemia reperfusion CG2: sham	Neutrophil infiltration, serum creatine kinase, serum and gastrocnemius muscle levels of protein carbonyl, xanthine oxidase, and adenosine deaminase.	CAPE reduced neutrophil infiltration and serum creatine kinase as well as protein carbonyl, xanthine oxidase, and adenosine deaminase levels in the blood and gastrocnemius muscle.	[102,103]
Gavage propolis (1 g/kg body weight/d/2 weeks)	Adult male Wistar rats with 2-week hind limb unloading (HU)	EG: HU rats + propolis CG1: normal rats + propolis CG2: normal rats CG3: untreated HU rats	Soleus muscle weight, FCSA, myofiber number, apoptosis of endothelial cells, capillary to muscle fiber ratio, capillary number, luminal diameter, and capillary volume, levels of ROS, SOD-1, anti-angiogenic factors, and pro-angiogenic factors.	Propolis had no effect on soleus muscle weight or FCSA. However, the relative soleus muscle-to-body weight and the capillary to muscle fiber ratio of the soleus muscle were significantly higher in EG than in CG3. Propolis decreased the number of apoptotic endothelial cells, improved levels of SOD-1, ROS, and VEGF leading to increased capillary number, luminal diameter, and capillary volume in the EG to the levels of CG1 and CG2, which were all significantly different from CG3.	[110]
Gavage propolis (200-mg/kg body weight/d/28 d)	Adult female Wistar rats undergoing crush injuries of the sciatic nerve	EG: propolis CG1: curcumin CG2: methylprednisolone CG3: sham rats CG4: untreated rats with sciatic nerve injury	Gastrocnemius muscle mass, motor function, nerve fiber myelination, and nerve conduction.	Propolis and curcumin significantly restored gastrocnemius muscle mass, improved walking, nerve fiber myelination, and motor conduction to the gastrocnemius muscle compared with CG4.	[113]
Dietary fresh monofloral bee pollen 5% or 10% (3 weeks)	Malnourished old male Wistar rats (22-month-old)	EG1 and EG2: refeeding diet + bee pollen 5% and 10%, respectively CG1: refeeding diet CG2: no treatment CG3: untreated normal weight rats	Body weight and composition, muscle mass, muscle protein synthesis rate, plasma cytokines, mitochondrial enzyme activity, and mTOR/p70S6kinase/4eBP1 signaling.	Bee pollen restored visceral and subcutaneous adipose tissues and increased plantaris and gastrocnemius muscle mass. 10% pollen restored the levels of cytokines to normal, boosted muscle protein synthesis, and increased complex IV activity while both 5% and 10% increased the activity CS and the phosphorylation of mTOR/p70S6kinase/4eBP1 signaling.	[36]
Oral crude and processed monofloral Indian mustard bee pollen (100, 200, and 300 mg/kg body weight/4 weeks)	Adult male Wistar rats and Swiss albino mice	EG1: neat bee pollen + acute eccentric swimming EG2: processed bee pollen + acute eccentric swimming CG1: no treatment CG2: bee pollen only CG3: acute eccentric swimming + vehicle gum acacia	Body weight, relative weight of the gastrocnemius muscle, SOD, GSH, MDA, NO, total protein content, lipid peroxidation, myostatin mRNA, β-actin, mitochondrial complex I, II, III, and IV enzyme activity.	Crude (300 mg/kg) and processed (200 and 300 mg/kg) bee pollen prevented myofiber fragmentation and restored body weight and the relative weight of the gastrocnemius muscle as well as mitochondrial complex-I, -II, -III, and -IV enzyme activity to normal (CG1 and CG2) compared with CG3. Both bee pollen treatments decreased MDA, NO, total protein content, lipid peroxidation, and myostatin and increased SOD and GSH in skeletal muscle.	[90]

N.B. All studies were conducted in vivo unless otherwise indicated. pRJ: protease-treated royal jelly, No: number, SC: Satellite cells, EG: experimental group, CG: control group, min: minute, d: day, h: hour, IGF-1: Insulin growth factor 1, AKT: Serine/threonine protein kinase, AS160: AKT substrate of 160 kDa, BALB/c: Bagg albino, 10-HDA: 10-Hydroxy-decanoic acid, ROS: reactive oxygen species, HET: genetically heterogeneous mice, AST: aspartate aminotransferase, ALT: alanine aminotransferase, T-CHO: total cholesterol, TLR4: toll-like receptors 4, TG: triglyceride, HFD: high fat diet, LPS: lipopolysaccharide, HDL: high density lipoprotein, LDL: low density lipoprotein, HOMA-IR: homeostatic model assessment of insulin resistance, PUFA: polyunsaturated fatty acids, TNFR1: tumor necrosis factor receptor 1, HU: hind limb unloading, IKK: IκB kinase, DMSO: dimethyle sulfoxoide, AICAR: 5-aminoimidazole-4-carboxamide ribonucleoside, CAPE: caffeic acid phenethyl ester, AGEs: advanced glycation end products, MMP-12: metalloproteinase-12, MyoD: myogenic differentiation 1; VEGF: vascular endothelial growth factor, NF-κB: nuclear factor kappa B, MDA: malondialdehyde, SOD: superoxide dismutase, CAT: catalase, and GPX: glutathione peroxidase, LIF: leukemia inhibitory factor, CXCL-10: chemokine (C-X-C motif) ligand 10, CCL: C-C chemokine ligand, MCP-1: monocyte chemotactic protein-1, COX-2: cyclooxygenase-2, iNOS: inducible nitric oxide synthase, PARP: poly (ADP-ribose) polymerase, AMPK: adenosine monophosphate activated protein kinase, PI3K: phosphatidylinositol 3-kinase, IR: insulin receptor, GLUT4: glucose transporter 4, OGTT: Oral Glucose Tolerance Test, FCSA: fiber cross-sectional area, mTOR: mammalian target of rapamycin, p70S6kinase: P70 ribosomal proteins S6 kinase, 4eBP1: eukaryotic translation initiation 4E-binding protein 1, SOD: superoxide dismutase, GSH: glutathione, MDA: malonaldehyde, NO: nitric oxide, MGO: methylglyoxal, EDL: extensor digitorum longus, IL-6: interleukin-6, IL-1β: interleukin-1β, TLR4: toll-like receptor 4, CS: citrate synthase, β-HAD: β-hydroxyacyl coenzyme A dehydrogenase.

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
