# Peer review of "Apitherapy for Age-Related Skeletal Muscle Dysfunction (Sarcopenia): A Review on the Effects of Royal Jelly, Propolis, and Bee Pollen"

_foods, 2020, doi:10.3390/foods9101362_

Round 1

Reviewer 1 Report

Comments to authors:

The article is well written and provides a comprehensive review of the topic

Minor comments:

  1. Line 20

“Not a single drug has been approved for sarcopenia treatment until the current moment, although it has multiple detrimental  effects including falls, hospitalization, disability, and institutionalization”

The words “until the current moment” may not be necessary.

It will be beneficial if the authors can list a few pharmacological agents that are currently in different stages of trials or being explored for sarcopenia such as-  Bimagrumab, Sarconeos, and exercise mimetics that are shown to induce muscle biogenesis.

  1. Line 24

 “Royal jelly, bee pollen, and propolis are common bee products that are rich in highly potent antioxidants such as flavonoids, phenols, amino acids, etc”.

“ etc” must be removed

  1. Lines 131-133

“drug-resistant bacterial strains such as S. aureus MRSA, P. aeruginosa, Klebsiella pneumoniae, enterococci (VRE), extended-spectrum β-lactamase-producing, Proteus mirabilis, and Escherichia coli”

The names of the bacterium must be consistent i.e Methicillin resistant Staphylococcus aureus

It would be better to have them spelled such as Klebsiella pneumonia instead of S. aureus and P. aeruginosa (Pseudomonas aeruginosa- is preferred)

  1. Line 159

Royal jelly has been historically used as a beautifying agent by famous queens such as Cleopatra, and it is still involved in cosmetic industry nowadays

 “Still involved” and “nowadays” – appear to be redundant. Please rewrite the sentence.

  1. Line 166

addition, bee queens (which enjoy long lifespan as well as super fertility and physical qualities)  consume royal jelly all over their entire lives

The sentence can be changed to “consume royal jelly throughout their lives”

No major comments

Author Response

Manuscript ID: foods-931293

Apitherapy for age-related skeletal muscle dysfunction (sarcopenia): A review on the effects of royal jelly, propolis, and bee pollen

Response to Comments of Reviewer 1

We thank reviewer 1 for his/her productive and insightful comments. The comments are addressed line-by-line as shown below. Replies come underneath in red.

Comment 1: Line 20: “Not a single drug has been approved for sarcopenia treatment until the current moment, although it has multiple detrimental  effects including falls, hospitalization, disability, and institutionalization”. The words “until the current moment” may not be necessary.

Response to Comment 1: Yes, “until the current moment” has been removed in this version (line 20-22).

Comment 2: It will be beneficial if the authors can list a few pharmacological agents that are currently in different stages of trials or being explored for sarcopenia such as-  Bimagrumab, Sarconeos, and exercise mimetics that are shown to induce muscle biogenesis. 

Response to Comment 2: Examples of experimental pharmacological agents have been integrated into the text as the reviewer indicated (line 21, 70-72).

Comment 3: Line 24:  “Royal jelly, bee pollen, and propolis are common bee products that are rich in highly potent antioxidants such as flavonoids, phenols, amino acids, etc”.  “ etc” must be removed

Response to Comment 3: Yes, “etc” has been removed, and the sentence was modified accordingly (line 25).

Comment 4: Lines 131-133: “drug-resistant bacterial strains such as S. aureus MRSA, P. aeruginosa, Klebsiella pneumoniae, enterococci (VRE), extended-spectrum β-lactamase-producing, Proteus mirabilis, and Escherichia coli”. The names of the bacterium must be consistent i.e Methicillin resistant Staphylococcus aureus. It would be better to have them spelled such as Klebsiella pneumonia instead of S. aureus and P. aeruginosa (Pseudomonas aeruginosa- is preferred)

 Response to Comment 4: Yes, thank you very much. The names of bacteria have been consistently spelt out (line 130-132).

Comment 5: Line 159: Royal jelly has been historically used as a beautifying agent by famous queens such as Cleopatra, and it is still involved in cosmetic industry nowadays

 “Still involved” and “nowadays” – appear to be redundant. Please rewrite the sentence.

Response to Comment 5: Yes, the redundant word “nowadays” has been removed (line 158).

Comment 6: Line 166: addition, bee queens (which enjoy long lifespan as well as super fertility and physical qualities)  consume royal jelly all over their entire lives.  The sentence can be changed to “consume royal jelly throughout their lives”

Response to Comment 6: Yes, the sentence has been changed to “consume royal jelly throughout their lives” (line 165).

In final, we thank reviewer 1 for the time, effort, and help provided. We hope that the comments were properly handled and that the revised version will be suitable for publication.

 Best regards,

Reviewer 2 Report

The present work addresses an original topic, of a great interest due it’s the contribution in gathering and providing scientific information regarding the effect of bee products (RJ, propolis and pollen) in age-related skeletal muscle dysfunction (sarcopenia) in a clear manner, but there is minor revision of the manuscript to be considered before further publication.

  • The key words are too long condensed and not readable, a short and clear key words can fulfill the purpose. 
  • line 124 needs referece
  • Bacterial species in italic
  • Lines 180-181:  the authors gives values of most abundant component of propolis 7-isopentenyloxucoumarin, boropinic acid, 4-geranyloxyferulic acid,  I think it’s for a specific propolis not propolis in general, as far as we know, up today there is no standardization of its chemical composition all over the world.
  • Lines 199-202: same remarks on component values
  • Section 2.2.: authors did not mention any information about the volatile part of propolis, although it present a high importance, it contains several bioactive compounds with a high spectrum of biological activities which could be involved on allevating / tereating age-related skeletal muscle dysfunction.
  • Standarize the citation of minerals, lines 158 and 227
  • It is worth to note that in the literature allergic reactions have been described in several case reports as an example the following paper : DOI: 10.1002/cbdv.201900094.
  • Figure 1 needs references. 
  • Line 378: separate RJ effect in an individual paragraph
  • For a best understanding of this section, keep same order of organisation when citing the three bee products, RJ-Propolis and pollen instead propolis pollen RJ. 
  • Table 1: is better to make the in vitro assays first then mention in vivo and redraw the table presentation to have clearer view.
  • Standarize the writing of the term royal jelly in whole manuscript, we find RJ, Royal jelly and royal jelly.
  • Lines 467-488 a text limitation is necessary with/without schema which can be useful for the explanation of the metabolic regulation.
  • Lines 539-554 this paragraphs can be shorter, it did not serve the context of the review, it is a repetition of what we can find in the literature.
  • Lignes 1034 and 1339: 1990/1995 too old as references.

Author Response

Manuscript ID: foods-931293

Apitherapy for age-related skeletal muscle dysfunction (sarcopenia): A review on the effects of royal jelly, propolis, and bee pollen

Response to Comments of Reviewer 2

We appreciate Reviewer 2’s thoughtful reading and concerns for clarity as indicated by the provided comments. The comments are addressed line-by-line as shown below. Replies come underneath in red.

 Comment 1: The key words are too long condensed and not readable, a short and clear key words can fulfill the purpose. 

Response to Comment 1: Yes, we have shortened and simplified the key words used in this revision (line 43-45)

Comment 2: line 124 needs referece –“Protein is the most copious active element in royal jelly representing half the weight of its dry matter”

Response to Comment 2: A relevant reference has been added to the indicated sentence (line 123).

Comment 3: Bacterial species in italic

Response to Comment 3: Yes, thank you. Bacterial species are changed in italic face (line 130-132).

Comment 4: Lines 180-181:  the authors gives values of most abundant component of propolis 7-isopentenyloxucoumarin, boropinic acid, 4-geranyloxyferulic acid,  I think it’s for a specific propolis not propolis in general, as far as we know, up today there is no standardization of its chemical composition all over the world.

Response to Comment 4: We agree with the reviewer, the chemical composition of most bee products has not been standardized until now. Referring to the original paper, we indicated the type of propolis (Italian) in which these compounds were characterized (line 179-181).

Comment 5: Lines 199-202: same remarks on component values

Response to Comment 5: Yes, we have indicated the type of propolis (poplar propolis found in Spain, line 199) wherein pinocembrin exists in high amounts.

Comment 6: Section 2.2.: authors did not mention any information about the volatile part of propolis, although it present a high importance, it contains several bioactive compounds with a high spectrum of biological activities which could be involved on allevating / tereating age-related skeletal muscle dysfunction.

Response to Comment 6: Thank you so much for helping us expand the functionality of the material of this review. The text has been modified to shed the light on the volatile compounds of propolis by adding a complete paragraph on this topic (line 202-214).

Comment 7: Standarize the citation of minerals, lines 158 and 227.

Response to Comment 7: Thank you for this comment. We have standardized the description of minerals (line 149-150,188-189)

Comment 8: It is worth to note that in the literature allergic reactions have been described in several case reports as an example the following paper : DOI: 10.1002/cbdv.201900094.

Response to Comment 8: Thank you so much for the resources provided by the reviewer—we have included them in the revision, really helpful.

Comment 9: Figure 1 needs references. 

Response to Comment 9: The caption of Figure 1 has been supplemented with the relevant references as the reviewer required (Line 340-359).

Comment 10: Line 378: separate RJ effect in an individual paragraph

Response to Comment 10: Effects of RJ were separated in an individual paragraph (Line 397).

Comment 11: For a best understanding of this section, keep same order of organization when citing the three bee products, RJ-Propolis and pollen instead propolis pollen RJ. 

Response to Comment 11: Yes, thank you so much. Of course, organization would be better if the order of all reports follows “RJ-Propolis and pollen sequence”, which the reviewer indicated. We have tried our best to stick to this order starting from the title and abstract in this revised version. However, it was not possible to stick to it in some instances given inconsistency of reported findings in reviewed studies, which may interrupt the flow of the overall idea.

Comment 12: Table 1: is better to make the in vitro assays first then mention in vivo and redraw the table presentation to have clearer view.

Response to Comment 12: We agree with the reviewer; starting Table 1 with in vitro assays, and then extending it to include in vivo studies may be the best presentation. We rearranged the sequence of studies accordingly. However, some studies comprised both in vitro and in vivo investigations, and in most instances, outcome indices were assessed both in vitro and in vivo. So, to avoid redundancy, these studies were reported with in vitro investigations on top of the in vivo ones. In order to keep the “RJ-Propolis and pollen sequence” noted in your former comment, in vitro investigations involving each bee product were moved on top of in vivo investigations involving the same product (Table 1, line 407).

Comment 13: Standarize the writing of the term royal jelly in whole manuscript, we find RJ, Royal jelly and royal jelly.

Response to Comment 13: That is right. In this revised version “royal jelly” is fully spelt, and its abbreviation is used only with protease treatment “pRJ”, which is defined on the list of abbreviations.

Comment 14: Lines 467-488 a text limitation is necessary with/without schema which can be useful for the explanation of the metabolic regulation.

Response to Comment 14: We thank the reviewer for such concerns about clarity. Yes, we have rewritten this section to make it briefer and clearer (line 486-514), we hope that it is now up to the standard that the reviewer would accept.

Comment 15: Lines 539-554 this paragraphs can be shorter, it did not serve the context of the review, it is a repetition of what we can find in the literature.

Response to Comment 15: We agree that information provided in this paragraph is readily available in the literature, albeit a bit of information may pave the way for noted results and clarify our point. According to this comment, we have edited this paragraph and made it a bit shorter by removing redundant statements (line 556-569).

Comment 16: Lignes 1034 and 1339: 1990/1995 too old as references.

Response to Comment 16: We have replaced old references with new ones, and modified the text accordingly (line 256, 614-616).

We hope that the comments were properly handled and that the revised version will be suitable for publication.

 Best regards,
